# A Temporal Graph Learning Framework for Lead-Lag Detection in Financial Markets

## Abstract

Lead-lag relationships and effects among financial assets are fundamental for understanding market dynamics and predicting price movements. However, accurately detecting these evolving temporal dependencies remains a complex challenge. Traditional approaches predominantly rely on statistical methods based on price evidence, while machine learning and deep learning techniques remain largely unexplored in this context. The lead-lag relationships and effects can be naturally represented using a dynamic graph structure, although this direction is still uninvestigated in the literature. Indeed, existing studies rarely leverage graph-based representations, and when they do, they typically consider static rather than dynamic structures, limiting their ability to capture temporal evolution. To overcome these limitations, this study proposes a novel framework that: (*i*) formulates lead-lag relationships and effects detection as a temporal link prediction task on dynamic graphs; (*ii*) introduces a novel real-world benchmark task for the evaluation and comparison of TGNNs; (*iii*) adapts, extends, and defines eight deep learning models ranging from simple LSTMs to State-of-the-Art Temporal Graph Neural Networks (TGNNs); (*iv*) explicitly evaluates two scenarios: lead-lag relationships that are both positive and negative, as well as those that are only positive; (*v*) performs an ablation study to assess the impact of the key components of the considered approaches. The experiments were conducted on a custom-gathered dataset of financial assets enriched with temporal, structural, and sentiment features. The findings demonstrate that temporal graph learning effectively models complex lead-lag relationships, opening new avenues for data-driven financial market analysis.

## 1 Introduction

A *lead-lag relationship* refers to a pattern where changes in one financial asset happen just before changes in another, usually over short time intervals. These relationships often arise frequently and may not hold statistical significance. In contrast, a *lead-lag effect* is a more robust and consistent phenomenon, where the price movement of one asset reliably follows another after a longer period, suggesting a stronger causal link. In this scenario, the effects are generally identified by the systematic occurrence of lead-lag relationships between the two assets over time (Li et al., 2022). Figure 1 demonstrates the lead-lag concept, with the orange line representing the leading asset, whose price changes occur before those of the blue line, the lagging asset. In a real-world scenario, the orange asset could represent raw food materials, while the blue line could represent the price of cooked food, demonstrating how changes in raw food costs precede changes in cooked food prices. Over an extended period, the consistent occurrence of lead-lag relationships may support the existence of a lead-lag effect between the leading and lagging assets.

Understanding lead-lag relationships and effects between financial assets is a long-standing problem rooted in the history of economic exchange that is particularly valuable for informing trading strategies and risk management. Yet, detecting such patterns remains challenging. Most existing approaches rely on statistical methods or direct price prediction, often overlooking the complex time and structure interdependencies between assets (Scherbina & Schlusche, 2020; Li et al., 2022). Furthermore, there is a lack of data-driven solutions and minimal use of machine learning (ML) and deep learning (DL) techniques. One reason for this is the complexity of the data typically involved in these kinds of tasks. Indeed, lead-lag relationships involve interactions and dependencies

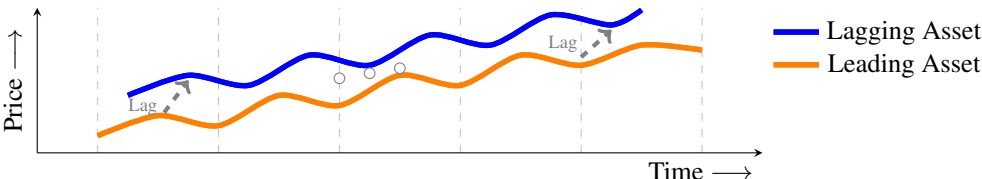

Figure 1: Plot illustrating the lead-lag relationship between two assets, with the orange line leading and the blue line lagging over time. The consistent occurrence may support the existence of a lead-lag effect between the two.

that change over time between financial assets, which can be naturally represented as a dynamic graph. The graph structure effectively models the evolving connections, making it a powerful tool for capturing and analyzing these temporal patterns. Recent developments in Graph Neural Network (GNN) architectures (Hou et al., 2022; Dwivedi & Bresson, 2020; Pasa et al., 2024), particularly those designed for temporal graph data (Longa et al., 2023; Cong et al., 2023; Jin et al., 2024), have made dynamic graph models especially promising. As a result, exploring graph-based methods for modelling lead-lag relationships is both timely and necessary. Additionally, the task is particularly interesting because its inherent complexity and the importance of dynamic asset interactions make lead–lag detection a powerful benchmark for evaluating DL models on temporal graph data.

This work introduces a novel formulation of the lead–lag detection problem, framing it as a temporal link prediction task on dynamic graphs, where the graph topology evolves over time. In this formulation, assets are represented as nodes, and directed temporal edges capture their predictive influence over time. Building on this representation, this study proposes a Graph Neural Network (GNN)–based framework (Sperduti & Starita, 1997; Scarselli et al., 2009; Micheli, 2009) capable of uncovering complex non-linear patterns. Specifically, the dynamic nature of the problem makes it well-suited for Temporal Graph Neural Networks (TGNNs) (Longa et al., 2023; Jin et al., 2024), as it offers a promising approach to capturing both temporal and structural dependencies in financial networks. In contrast to prior research on lead-lag detection and pricing trends, which often emphasizes pairwise (Shi et al., 2023) or static analysis (Shi et al., 2024), the proposed approach enables the simultaneous modelling of multiple assets with interdependent time dynamics. While this new formulation facilitates the modeling of intricate, time-evolving lead-lag relationships, it inherently precludes direct comparisons with traditional non-ML methodologies currently prevalent in the field. Overall, the framework offers a new perspective on lead–lag detection while extending the problem beyond the limits of traditional sequential approaches and statistical methods.

To support the proposed approach, this study introduces a custom-gathered dataset[1] comprising stocks and commodities, enriched with five years of daily pricing data, financial indicators, and sentiment features. To assess the boundaries of this approach to the lead-lag detection problem, this study adapts, develops, and evaluates several solutions based on multiple State-of-the-Art temporal GNN architectures, including JODIE (Kumar et al., 2019), DySAT (Sankar et al., 2020), TGAT (Xu et al., 2020), TGN (Rossi et al., 2020), APAN (Wang et al., 2021), and GraphMixer (GM) (Cong et al., 2023), a simple yet effective architecture based entirely on multilayer perceptrons (MLPs). Despite its simplicity, GM outperforms all other models in capturing lead-lag relationships, demonstrating its ability to effectively model temporal and structural dependencies in financial networks, supporting the evidence presented by Cong et al. (2023). Each model was evaluated in two different scenarios: one considering lead-lag relationships that are both positive and negative, and the other only positive. Additionally, an in-depth analysis of each model is conducted to understand the contribution of the approaches' key components. The results highlight both the benefits of leveraging a temporal graph framework and the contribution of a novel real-world benchmark task for the evaluation and comparison of TGNNs.

To summarize, the contributions of this work include: (*i*) the redefinition of the detection of lead-lag relationships and effects as a temporal link prediction task on dynamic graphs; (*ii*) the introduction of a novel task for temporal GNNs, leveraging models specifically designed for structured domains; (*iii*) A novel benchmark dataset of stocks and commodities, with five years of daily prices, financial

---

[1]The dataset is included as Supplementary Material and will be made available upon the paper's acceptance.

indicators, and sentiment features. *(iv)* the adaptation, extension, and definition of eight DL models, ranging from simple LSTMs to State-of-the-Art temporal graph learning models; *(v)* the explicit evaluation of two scenarios: one considering lead-lag relationships that are both positive and negative, and the other only positive. *(vi)* the conduct of an ablation study to assess the impact of the key components of the considered approaches.

## 2 BACKGROUND AND RELATED WORKS

### 2.1 LEAD-LAG DETECTION

Despite the practical significance of the lead-lag phenomenon, the problem is not well-documented in the literature, with a lack of formal definitions. For instance, the terms *lead-lag effect* and *lead-lag relationship* are often used interchangeably (Li et al., 2022; Basnarkov et al., 2020). However, a more precise definition, which is also utilized in this work, is derived from the efforts of Li et al. (2022). Furthermore, the literature does not specify whether lead-lag effects should solely represent positive relationships, such as increases in an asset's price, or include both positive and negative relationships, encompassing both rises and falls in asset prices. This study considers both scenarios.

Building on the distinction between lead-lag relationship and the lead-lag effect, existing literature offers various methods for detecting and analyzing these phenomena. Traditional approaches often utilize high-frequency data, like five-minute intervals, to leverage short-term imbalances in order flow and liquidity dislocations, aiming to identify transient lead-lag relationships (Scherbina & Schlusche, 2020), though these relationships may not persist or hold statistical significance over longer periods (Li et al., 2021). Conversely, studies focusing on lower-frequency data propose more robust methodologies for identifying stable lead-lag effects that consistently manifest over time. Some studies build daily lead-lag networks by checking whether one asset's return aligns with another's previous-day return and aggregating these instances for significance (Li et al., 2022). This method eases the detection of structural lead-lag effects that may result from sectoral dependencies, macroeconomic influences, or systematic liquidity effects. Additionally, while short-term lead-lag relationships are primarily influenced by market microstructure effects, longer-term lead-lag effects provide a stronger theoretical basis for portfolio optimization and systematic trading strategies, offering valuable insights into market inefficiencies (Li et al., 2022).

The literature about lead–lag effects has only superficially explored machine learning applications. Han & Kong (2022) propose a method to study the lead-lag relations in commodity futures returns based on sparse feature selection via least absolute shrinkage and selection operator (LASSO). In the context of graph-based approaches, Li et al. (2024) model cross-asset lead–lag relationships using static graphs as inputs to predict future stock movements. These approaches tend to overlook the valuable relational information available through network-based models, potentially missing the interconnected nature of financial assets of graph-based approaches. To the best of our knowledge, no GNN or TGNN-based methodology has yet been applied to lead–lag detection.

### 2.2 GRAPH NEURAL NETWORKS

A graph is defined as $G = (V, E, L)$, where $V = \{v_i\}_{i=1}^n$ denotes the set of $n$ nodes (or vertices) of the graph, $E \subseteq V \times V$ is the set of edges connecting the nodes, and $L = \{l_i\}_{i=1}^n$, with $l_i \in \mathbb{R}^s$ is the set of $s$ attributes (or features) associated to each vertex $v_i \in V$. In the following, we adopt the notation conventions presented in Appendix A. Let $\mathcal{N}(v_i)$ be the set of nodes adjacent to $v_i$. A Graph Neural Network (GNN) is a model that exploits the structure of the graph and the information embedded in feature vectors of each node in order to learn a representation $h_v \in \mathbb{R}^m$ for each vertex $v \in V$. In modern GNNs models, the computation of $h_v$ can be divided in two main steps, *aggregate* and *combine*: $h_v = \text{Comb}(l_v, \text{Agg}(\{l_u : u \in \mathcal{N}(v)\}))$. It is possible to extend the range of the considered neighborhood by iteratively performing aggregation and combination for $k$ iterations. In this way, a hidden representation $h_v^{(k)}$ of the node $v$ is obtained, which contains information about the structure and the neighbours that are at distance $k$ from $v$:

$$h_v^{(i)} = \text{Comb}(h_v^{(i-1)}, \text{Agg}(\{h_u^{(i-1)} : u \in \mathcal{N}(v)\}));$$

where $h_v^{(0)} = l_v$, $\quad h_v^{(i)} \in \mathbb{R}^{m_i}$, $i \in [0, \ldots, k]$ and $m_i$ is the size of convolutional layer $i$. Thus, a deep GNN of $k$-layers is obtained. The choice of aggregation function and combination function

defines the type of Graph Convolution adopted by the GNN. For example, Micheli (2009) proposed the first model that uses graph convolutions, while Kipf & Welling (2017) derived a widely adopted formulation. Subsequent works have extended these foundations by introducing more expressive aggregation schemes (Hamilton et al., 2017; Xu et al., 2019), attention-based mechanisms (Velick­ovic et al., 2018), and message-passing frameworks (Gilmer et al., 2017), enabling GNNs to capture increasingly complex graph structures and node interactions.

## 2.3 TEMPORAL GRAPH NEURAL NETWORKS

While many graph applications involve static structures, real-world networks often exhibit tempo­ral dynamics in either their topology, node features, or both. These temporal dynamics introduce additional complexity to graph learning tasks. To address this, a variety of temporal graph neural networks (TGNNs) (Longa et al., 2023; Jin et al., 2024) have been proposed, which extend stan­dard GNNs to capture time-evolving information. To model temporal dynamics, the *static graph* definition needs to be extended to a *temporal graph*. A temporal graph is defined as a sequence of graphs over time: $G^{(t)} = (V^{(t)}, E^{(t)}, L^{(t)})$, where each $G^{(t)}$ represents the graph at a discrete time step $t \in \{1, \ldots, \tilde{t}\}$. Early work, such as JODIE (Kumar et al., 2019), is based on the application of a RNNs approach designed for learning an embedding trajectory in dynamic interaction networks. JODIE focuses on modelling the co-evolution of node embeddings by using coupled recurrent up­dates that allow user and item representations to evolve in response to interactions. Building on the idea of temporal dependencies, DySAT (Sankar et al., 2020) introduces a snapshot-based ar­chitecture that leverages self-attention across both structural and temporal dimensions, enabling the model to learn patterns over sequences of graph snapshots. Unlike RNN-based models, DySAT's self-attention mechanism enables parallel computation and better scalability. An enhancement of the standard GAT (Velickovic et al., 2018) networks for temporal networks is proposed by Xu et al. (2020), which introduces the TGAT architecture. TGAT operates in a continuous-time setting and employs functional time encoding combined with temporal attention, allowing it to handle asyn­chronous events and generalize to unseen nodes in an inductive fashion. Expanding on continuous-time modelling, TGN (Rossi et al., 2020) is engineered to model dynamic graphs in which events oc­cur throughout continuous time. The framework employs an encoder-decoder architecture, wherein the encoder maps dynamic graph events into node embeddings through RNNs to encapsulate tem­poral information. TGN incorporates a memory-based architecture that stores historical states for each node and updates them using learnable message and memory functions, making it well-suited for event-driven dynamic graphs. APAN (Wang et al., 2021) further enhances temporal context modelling by using attentive pooling mechanisms that integrate both recent and long-term temporal information, capturing dependencies across a wide temporal window. The pooling mechanism aims to mitigate the high inference times imposed by other temporal GNN architectures dealing with tem­poral link prediction, since there are tasks, like financial fraud detection, that require extremely high inference speed in order for the models to be usable in real-world scenarios. Most recently, Graph-Mixer (Cong et al., 2023) shifted from attention-based designs to a token-mixing strategy inspired by MLP-Mixer (Tolstikhin et al., 2021) architectures, achieving temporal and structural fusion via lightweight mixing operations, enhancing scalability while maintaining representational power.

## 3 METHODOLOGY: PROPOSED FRAMEWORK

### 3.1 PROBLEM FORMULATION

Given the definition of lead-lag relationships and lead-lag effects, this work addresses the detec­tion of *lead-lag patterns* by lessening the distinction between relationships and effects, aiming to model consistent lead-lag effects while also capturing occasional lead-lag relationships. The aim is to model the dynamics of lead-lag pattern detection through temporal graph link prediction, where nodes represent assets, e.g., stocks and commodities. A directed link from asset $j$ to asset $i$ is estab­lished when lagging asset $i$ follows leading asset $j$ at a specific time step $t$. The ambiguous lead-lag definitions in current literature introduce uncertainty regarding whether observed relationships and effects are exclusive to positive price movements, necessitating an increase in price, or encompass any price movement, allowing for fluctuations in either direction. For this reason, this study will investigate both definitions.

Mathematically, let $p_i^t$ and $r_i^t$ denote the *closing price* and the *percentage return*, respectively, of asset $i$ on day $t$, where $r_i^t = \frac{p_i^t - p_i^{t-1}}{p_i^{t-1}} \times 100$. Then, given a minimum threshold $\epsilon > 0$, a lead-lag relationship between asset $j$ and asset $i$ is defined if the return of the leader asset $j$ at day $t-1$ and the return of the follower asset $i$ at day $t$ both exceed this threshold in the same direction (positive or negative). Formally, the lead-lag relationship is established when the following condition holds:

$$
\begin{cases}
r_j^{t-1} \geq \epsilon \text{ and } & r_i^t \geq \epsilon, \quad \text{if} \quad r_j^{t-1}, r_i^t > 0; \\
-r_j^{t-1} \geq \epsilon \text{ and } -r_i^t \geq \epsilon, & \quad \text{if} \quad r_j^{t-1}, r_i^t < 0.
\end{cases}
\tag{1}
$$

In this formulation, unlike other bound-based approaches (Li et al., 2022), there is no requirement for the follower's return to match the leader's return in magnitude, only that both exceed the absolute threshold $\epsilon$ in the same direction.

**Problem Formulation and Statistical Finance Methods**  The statistical methods typically examine pairwise relationships in isolation, while the framework proposed in this paper captures the full network of interdependencies among multiple assets simultaneously. The threshold-based relationship definition in Equation 1 represents a paradigmatic shift from traditional statistical approaches. Classical methods like *Granger causality* focus on linear predictive relationships, whereas the proposed formulation establishes lead-lag relationships based on directional price movements exceeding absolute thresholds. While it would be theoretically possible to define extensions of traditional statistical methods to allow comparison with the proposed dynamic GNN-based approach, these adaptations would essentially create hybrid approaches that differ substantially from the established statistical methods. Consequently, the development of adapted statistical models is a complex task that lies outside the scope of this study.

### 3.2 DATASET COLLECTION AND GRAPH CONSTRUCTION

The raw data was gathered via *company-name*[2] APIs from the New York Stock Exchange (NYSE) and the Chicago Mercantile Exchange Group (CME Group). Additionally, consistency checks ensured price precision, addressing missing values and market closures like weekends and holidays. Commodities were included alongside companies due to their mutual influence on price behaviour. A heuristic approach was used for selection, rather than broad indices like the S&P 500 (Blume & Edelen, 2004) or FTSE (Danbolt et al., 2018). Since the dataset aims to capture a wide range of financial data to identify patterns in asset prices, it includes historical prices, timestamps, financial indicators, sentiment, and textual data. Five sectors were identified: (*i*) energy (oil, gas, renewables); (*ii*) technology (semiconductors, microchips); (*iii*) materials (metals, mining, chemicals); (*iv*) automotive (traditional, electric vehicles); (*v*) industrials (batteries, energy storage). From these, 29 companies, including Tesla, NVIDIA, and Ford, and eight commodities like gold and crude oil, were chosen for a total of 37 entities. For each entity, daily features were extracted from `2019/06/17` to `2024/06/12` to support detailed analysis and learning. These features include typical financial metrics — close price, open price, high, low, change, percent change, and volume-weighted average price — and technical indicators. A detailed discussion of the dataset features is in Appendix B.

In addition to the conventional indicators often present in financial models, daily sentiment data was incorporated in the analysis, offering insight into its potential influence on lead-lag relationships. Ultimately, the data collection process was concluded by extracting a concise description for each asset in the dataset from an LLM (GPT-4o), which was subsequently embedded into 384-dimensional vectors using a pre-trained $paraphrase - MiniLM - L6 - v2$ sentence transformer from $huggingface$ (Reimers & Gurevych, 2019; Hurst et al., 2024).

Each asset is then represented as a node in the temporal graph, while edges represent lead-lag relationships: if at time $t$, the price of node $v_i$ changes by at least $\epsilon\%$, and at $t + \tau$ the price of node $v_j$ also changes by at least $\epsilon\%$, an edge exists from $v_i$ to $v_j$ at time $t$. According to Li et al. (2022), $\epsilon$ demonstrates robustness in lead-lag modeling, with minimal outcome variation when altered. Unlike the long-term focus in their study, this work also addresses short-term lead-lag interactions for investment insights. For this reason, a value of $\epsilon = 5\%$ is used to balance graph density, where lower values of $\epsilon$ lead to numerous random connections, while higher values result in sparse networks. Following the discussion presented in Sheth et al. (2023), this threshold maintains a balanced approach

---

[2]Anonymized for double-blind review.

that avoids the excessive trading frequency observed at lower thresholds, while also avoiding a more restrictive and conservative model, employing higher values, where the graph nodes rarely interact with each other. Considering the daily granularity of our data and given that the lead-lag effects are typically observed in higher-frequency data (Li et al., 2022), the optimal $\tau$ value, accordingly to the literature (Li et al., 2021; 2022; Sheth et al., 2023), is set to 1, indicating a more rapid and proactive investment strategy. More details on the graph statistics are reported in Appendix C.

### 3.3 SEQUENTIAL BASELINE METHOD

With the aim of providing a solid evaluation of the proposed graph-based model formulation, a sequence-only baseline is also introduced, which ignores the graph structure described in the previous sections. The sequential baseline employs a temporal LSTM architecture that fundamentally treats link prediction as an isolated sequence modelling problem, i.e., it does not consider inter-asset relationships. The model consists of three components: *(i)* a temporal sequence encoder using LSTM/GRU layers that processes the last $k$ historical edge features chronologically, *(ii)* a multi-layer perceptron that encodes current edge features, and *(iii)* a fusion network that concatenates these representations for binary link prediction via sigmoid activation. Training uses binary cross-entropy loss with a sophisticated negative sampling through node corruption, maintaining temporal consistency by ensuring validation and test splits can only access historical data from previous time steps. This approach exhibits structural blindness, predicting each edge in isolation and ignoring the concurrent network topology, effectively reducing a graph problem to simple time-series prediction.

### 3.4 TEMPORAL GRAPH-BASED MODELS

To assess the feasibility of performing lead-lag detection by modelling the problem using temporal graphs, several solutions based on State-of-the-Art models needed to be implemented. In particular, JODIE, DySAT, TGAT, TGN, APAN, and GraphMixer (GM) were considered, with each model serving as a foundation for developing a lead-lag detector tailored to the task tackled in this study. Adaptations began from the edge prediction variants of these models, as proposed by Cong et al. (2023). However, each architecture needed ad-hoc modifications to fit the lead-lag detection framework due to differences in modelling assumptions, encoding strategies, and interaction mechanisms.

The original JODIE model is designed for bipartite graphs and focuses on predicting future interactions between nodes of different types. For this study, it was adapted to a homogeneous setting by treating all nodes uniformly as assets (stocks or commodities), where each temporal edge $e_{ij}^{(t)}$ denotes a lead-lag relationship. Each node maintains a dynamic embedding that evolves over time as interactions occur. In the new adaptation, when asset $v_i$ leads asset $v_j$ at time $t$, their temporal embeddings are updated using a custom RNN-based encoder that takes into account the elapsed time since their last interactions and the features of the edge. These updated embeddings are then passed to a decoder network that predicts the likelihood of a lead-lag relationship at the current time step. DySAT combines structural self-attention and temporal self-attention over a sequence of static graph snapshots. For lead-lag detection, each snapshot is interpreted as a windowed observation of dynamic pairwise lead-lag effects, using a binary adjacency matrix to encode whether $v_i$ leads $v_j$ within each window. The structural attention is computed for each snapshot, while temporal attention aggregates information across time, enabling the model to encode persistent lead-lag patterns. TGAT uses temporal self-attention with continuous-time encoding. It is adapted by representing interactions as timestamped edges and assigning temporal encodings to both node embeddings and the edge itself. Each attention head processes temporal proximity and feature similarity to detect directional dependencies. TGN maintains a memory for each node, updated upon interactions. Lead-lag relationships are treated as directed interactions that update both source and target memories using message functions based on node and edge features, and retrieve embeddings via a memory-based projection. APAN employs adaptive path aggregation across temporal sequences of node pairs. Paths are constructed from potential leader to lagger assets and aggregate information through time-aware attention mechanisms, capturing indirect lead-lag chains. Lastly, GM integrates temporal and topological mixing using stacked layers of permutation-invariant MLPs. It was adapted to lead-lag detection by organizing temporal edges into sequences $\{e_{ij}^{(t_k)}\}_{k=1}^{\tilde{t}}$, where each $e_{ij}^{(t_k)}$ indicates that asset $v_i$ leads $v_j$ at time $t_k$, and $\tilde{t}$ represents the final time step in the considered period. Each edge is encoded as a tuple $(\mathbf{l}_i, \mathbf{l}_j, t_k) \in \mathbb{R}^d$, and passed through a time-mixing layer followed by

a node-mixing layer. Formally, given an input tensor $\mathbf{Z} \in \mathbb{R}^{\tilde{t} \times d}$ of temporal edge features, the time-mixing transformation is defined as $\mathbf{Z}' = \mathrm{MLP}_{\mathrm{time}}\left(\mathbf{Z} + \mathrm{PE}(t_1, \ldots, t_{\tilde{t}})\right)$, where $\mathrm{PE}(\cdot)$ is a positional encoding of time step. This is followed by a node-wise permutation-invariant aggregation $\mathbf{h}_{ij} = \mathrm{MLP}_{\mathrm{node}}\left(||\mathbf{Z}'^{(k)}_{i,j}||^{\tilde{t}}_{k=1}\right)$, producing the final edge representation $\mathbf{h}_{ij}$, which is then used to predict the probability of a lead-lag link $p_{ij} = \sigma\left(\mathbf{w}^{\top}\mathbf{h}_{ij} + b\right)$, where $\sigma$ is the sigmoid function and $\mathbf{w}, b$ are learnable parameters. In all models, the prediction task is recast as a binary classification problem over directed temporal edges, where a positive label indicates that $v_i$ leads $v_j$ at time $t$.

**GraphMixer-TNF (GM-TNF)** The temporal graph depicting a lead-lag pattern in stocks and commodities forms a highly dynamic system, potentially necessitating nodes with time-varying features. Disregarding the ongoing changes in node attributes that represent such dynamic entities can result in a suboptimal model. Therefore, a new version of GM, dubbed **GraphMixer-Temporal Node Features (GM-TNF)** has been developed. This version incorporates a node encoder capable of managing the fluctuating dynamics of financial data. Importantly, the core concept remains unchanged, with the sole adjustment being in the computation of the node representation at time $t_0$, defined as $\boldsymbol{l}_i^{t_0} = \boldsymbol{l}_i^{t_1} + \mathrm{Mean}\left\{\boldsymbol{l}_j^{t_1} \mid v_j \in \mathcal{N}(v_i; t_0 - \delta, t_0)\right\}$, where $\delta$ is a hyperparameter denoting the number of time steps in the past considered for node aggregation, and $\mathcal{N}(v_i; t_j, t_h)$ is the set of the 1-hop neighborhood of node $v_i$ with time steps from $t_j$ to $t_h$. Each vector $\boldsymbol{l}_i^{t_1}$ represents the attributes of node $v_i$ at the last observed time step. Bear in mind that GM-TNF equals GM when both do not use temporal features as link attributes, e.g., closing prices.

## 4 EXPERIMENTAL ASSESSMENT

### 4.1 DATASET AND METRICS

Since the lead-lag problem lacks formal definitions regarding whether it should consider only positive relationships or both positive and negative ones, this study investigates both interpretations. For this reason, two versions of the dataset are generated following the process presented in Section 3. The first is a temporal graph where lead-lag relationships are defined considering both the conditions in Equation 1, i.e., $r_i^t > \epsilon$ and $-r_i^t < \epsilon$, while in the second version only the case in which $r_i^t > \epsilon$ is considered. To objectively assess the models' performances, this work adopts several key metrics already adopted in the State-of-the-Art (Cong et al., 2023), which collectively provide a thorough evaluation of the model's ability to rank and identify relevant temporal links. Additionally, each experiment is conducted *five times* to report both the average value and the standard deviation. A complete description of the considered metrics and their applications is reported in Appendix D.

Based on the attributes described in Section 3, the following feature groups are defined for both nodes and links: *(i)* Description Embeddings. Each node in the graph — i.e., a company or commodity — is represented with a 384-dimensional vector created by converting descriptions obtained from a language model into vector form. Link attributes are then composed by the embeddings of both source and destination nodes, i.e., a 768-dimensional vector. *(ii)* Embeddings + Prices. These features are composed of both the description embeddings and the closing price at time $t$. It results in 385-dimensional features for nodes and 770-dimensional features for links. Note that this is a time-dependent feature. *(iii)* Embeddings + Prices + Financial Indicators + Sentiment. These features incorporate precedents, along with financial indicators and sentiment scores, resulting in a 400-dimensional vector for nodes and an 800-dimensional vector for links.

### 4.2 MODEL SELECTION AND IMPLEMENTATION DETAILS

All models undergo fair model selection with training and validation sets, followed by evaluation of the chosen model on the test set. Specifically, the models are validated on the dataset considering both positive and negative lead-lag relationships, and then adopted "as-is" on the dataset made of only bullish trends. The experimental setup utilized in this study is adopted from Cong et al. (2023). Additionally, to facilitate an objective comparison, the TGL implementation provided by Zhou et al. (2022) is employed consistently across all evaluated models. Specifically, GM was adapted from its official repository to integrate with the TGL framework, ensuring a consistent comparison envi-

Table 1: Comparison when both positive and negative lead-lag relationships are considered.

| Model | Metrics | | | | | |
|---|---|---|---|---|---|---|
| | AP ↑ | AAUC ↑ | R@1 ↑ | R@5 ↑ | R@10 ↑ | MRR ↑ |
| LSTM | 0.51 ± 0.00 | 0.52 ± 0.00 | 0.07 ± 0.00 | 0.22 ± 0.00 | 0.38 ± 0.00 | 0.18 ± 0.00 |
| JODIE | 0.74 ± 0.03 | 0.77 ± 0.03 | 0.20 ± 0.03 | 0.58 ± 0.07 | 0.83 ± 0.06 | 0.30 ± 0.03 |
| DySAT | 0.73 ± 0.00 | 0.75 ± 0.02 | 0.25 ± 0.02 | 0.59 ± 0.02 | 0.77 ± 0.01 | 0.31 ± 0.00 |
| TGAT | 0.70 ± 0.03 | 0.73 ± 0.03 | 0.28 ± 0.00 | 0.57 ± 0.01 | 0.77 ± 0.00 | 0.33 ± 0.01 |
| TGN | 0.73 ± 0.02 | 0.75 ± 0.02 | 0.22 ± 0.02 | 0.58 ± 0.02 | 0.83 ± 0.02 | 0.31 ± 0.02 |
| APAN | 0.66 ± 0.05 | 0.70 ± 0.05 | 0.13 ± 0.01 | 0.43 ± 0.02 | 0.71 ± 0.02 | 0.24 ± 0.01 |
| GM-TNF | 0.75 ± 0.01 | 0.82 ± 0.01 | 0.38 ± 0.01 | 0.79 ± 0.00 | 0.95 ± 0.00 | 0.46 ± 0.00 |
| GM | **0.79 ± 0.01** | **0.85 ± 0.01** | **0.41 ± 0.02** | **0.86 ± 0.03** | **0.99 ± 0.01** | **0.47 ± 0.02** |

Table 2: Comparison when only positive lead-lag relationships are considered.

| Model | Metrics | | | | | |
|---|---|---|---|---|---|---|
| | AP ↑ | AAUC ↑ | R@1 ↑ | R@5 ↑ | R@10 ↑ | MRR ↑ |
| LSTM | 0.512 ± 0.008 | 0.508 ± 0.010 | 0.048 ± 0.026 | 0.192 ± 0.022 | 0.354 ± 0.041 | 0.154 ± 0.019 |
| JODIE | 0.718 ± 0.042 | 0.743 ± 0.038 | 0.174 ± 0.028 | 0.517 ± 0.026 | 0.787 ± 0.031 | 0.271 ± 0.021 |
| DySAT | 0.646 ± 0.022 | 0.679 ± 0.018 | 0.179 ± 0.064 | 0.425 ± 0.032 | 0.632 ± 0.032 | 0.249 ± 0.045 |
| TGAT | 0.673 ± 0.031 | 0.694 ± 0.024 | 0.149 ± 0.026 | 0.349 ± 0.029 | 0.606 ± 0.036 | 0.212 ± 0.013 |
| TGN | 0.621 ± 0.038 | 0.672 ± 0.036 | 0.139 ± 0.021 | 0.411 ± 0.054 | 0.710 ± 0.049 | 0.234 ± 0.023 |
| APAN | 0.572 ± 0.047 | 0.621 ± 0.052 | 0.044 ± 0.029 | 0.232 ± 0.103 | 0.497 ± 0.088 | 0.145 ± 0.040 |
| GM-TNF | 0.762 ± 0.007 | 0.823 ± 0.002 | 0.346 ± 0.001 | 0.832 ± 0.013 | 0.989 ± 0.001 | 0.435 ± 0.002 |
| GM | **0.791 ± 0.000** | **0.832 ± 0.000** | **0.436 ± 0.027** | **0.910 ± 0.027** | **0.996 ± 0.005** | **0.503 ± 0.017** |

ronment. Appendix E provides more information on Model Selection and Implementation Details, covering grid search values, final selected values, and hardware specifications for each model[3].

## 4.3 EMPIRICAL RESULTS

**Positive and Negative Relations**    Table 1 reports the results obtained when considering all the lead-lag relationships, i.e., both positive and negative. The reported results clearly demonstrate that all temporal graph-based models significantly outperform the sequence-only baseline (LSTM), highlighting the importance of incorporating relational structure into the modeling process. Among the graph-based approaches, GraphMixer (GM) attains the highest AP score of $0.79$, outperforming all other models. A notable strength of GM lies in its superior performance on recall metrics at various cutoffs: R@1, R@5, and R@10. These results suggest that GM retrieves relevant lead-lag interactions more effectively, making it a strong candidate for applications requiring precise link ranking. GM also achieves the highest MRR, with a score of $0.47$, indicating its ability to rank correct links more accurately on average. Additionally, it obtains the highest AAUC score ($0.85$), reflecting better model calibration and improved overall ranking stability. Table 1 also shows a decline across all indicators for GraphMixer-TNF compared to the standard GraphMixer, despite GraphMixer-TNF still being the second-best model overall across all metrics. When compared to GM, the results suggest that the additional temporal node features present in GM-TNF did not contribute meaningful extra information, which, indeed, can be captured by the temporal evolution of the topology in GM. From an investor's perspective, these results underscore GM's ability to uncover meaningful lead-lag relationships and effects between assets and predict future trends with high accuracy. Its strong performance in terms of Precision and AAUC values suggests its practical relevance for forecasting asset behavior, supporting more informed trading strategies. Furthermore, its low variability across metrics confirms the model's stability, a crucial trait for real-world applications.

**Only Positive Relationships**    The current analysis focuses solely on positive price changes.By excluding negative movements, this method isolates bullish momentum patterns, offering more actionable insights for investors seeking long (buy) opportunities. This selective filtering enhances the ability to anticipate upward trends and supports momentum-driven strategies and portfolio optimization. The results visible in Table 2 confirm the robustness of the graph-based models, particularly GM, despite the substantial reduction in the number of positive link samples due to the exclusion of negative price changes. Performance metrics remained relatively stable, indicating that the models successfully adjusted to the more constrained task and learned meaningful patterns under the revised data conditions. The model rankings remain consistent with previous evaluations. Once again, the LSTM model exhibits significantly lower performance than the graph-based models. GM

---

[3]Experimental code and data are in the Supplementary Material and will be released upon acceptance.

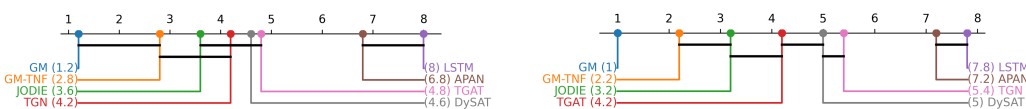

Figure 2: Critical difference diagrams of average score ranks: (left) both positive and negative relationships, and (right) only positive relationships.

Table 3: Impact of different types of features on model performance. For each model, the best AP value is highlighted in bold. Note that GM-TNF equals GM when both adopt only Embeddings.

| Feature Types | JODIE | DySAT | TGAT | TGN | APAN | GM-TNF | GM |
|---|---|---|---|---|---|---|---|
| Embeddings | **0.74 ± 0.03** | **0.73 ± 0.00** | 0.68 ± 0.02 | **0.73 ± 0.02** | **0.66 ± 0.05** | – | 0.78 ± 0.01 |
| + Prices | 0.68 ± 0.01 | 0.72 ± 0.02 | **0.70 ± 0.03** | 0.71 ± 0.02 | 0.64 ± 0.05 | **0.75 ± 0.01** | 0.77 ± 0.01 |
| + Fin. Ind. + Sent. | 0.69 ± 0.03 | 0.66 ± 0.03 | 0.69 ± 0.03 | 0.68 ± 0.02 | 0.62 ± 0.02 | 0.73 ± 0.02 | **0.79 ± 0.01** |

continues to perform best across most metrics. Additionally, improvements in the R@5 metric were observed for GM, with an increase of $0.05$. The most significant performance degradation was noted for APAN, which failed to identify any positive links at the top rank (R@1 $= 0.00$), further emphasizing the adaptability and stability of GM in capturing nuanced lead-lag patterns.

**Statistical Significance Analysis** To rigorously assess the importance of this study's findings, a statistical hypothesis test is conducted on all results reported. First, the Friedman test was applied to rank model accuracies per dataset run and evaluate the null hypothesis of model equivalence. This was followed by Conover's post-hoc test (Demsar, 2006), with results visualized in Figure 2. In both considered scenarios, the models exhibit statistically significant performance differences, with the GM models consistently outperforming all others. Conversely, the APAN and LSTM models are the least effective, failing to demonstrate any meaningful differences between the two approaches. A detailed discussion of these analyses is provided in Appendix F.

**Ablation Study** This section investigates the contribution of different feature types to each approach by using the dataset composed of positive and negative lead-lag relationships. Table 3 summarizes the performance of all the approaches according to the AP metric, while results for all other metrics can be found in Appendix G. Generally, the approaches perform best using only node description embeddings, without temporal-based features, such as prices. Exceptions are limited, like TGAT, which improves with the addition of price features, and GM, which excels only when all the features are used. This is consistent with the lead-lag graph construction, where temporal links reflect price fluctuations rather than exact price values, rendering explicit price features largely redundant. The addition of all the features tends to degrade the performance of most approaches, indicating that more complex temporal features may not always improve predictive capability in this domain. The same behaviour is observed even when considering the GM-TNF. For a more explicit comparison between GM and GM-TNF, refer to Appendix H.

## 5 Conclusions

This paper introduced a novel temporal graph neural network (TGNN) approach to tackle the lead–lag detection problem in financial networks. We proposed a new problem formulation that leverages temporal graph representations, allowing lead–lag detection to be naturally cast as a temporal link prediction task. To support this formulation, we constructed a custom dataset of financial assets and adapted, extended, and defined several State-of-the-Art TGNN-based methods, providing a reliable evaluation of the proposed framework. In addition, a purely sequential model was implemented as a baseline to assess the benefits of incorporating graph structure. Our empirical results demonstrate the clear advantages of using TGNNs for lead–lag detection. In particular, the GraphMixer model, despite its simplicity, outperforms all other approaches, effectively modelling both temporal and structural dependencies in financial networks. These findings not only validate the feasibility of the proposed approach but also introduce a novel real-world benchmark task for evaluating TGNNs on temporal financial graphs.

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

## A  NOTATION

The following mathematical notations are adopted: (*i*) lower case symbols for scalars and indexes, e.g., $a$; (*ii*) italics upper case symbols for sets, e.g., $A$; (*iii*) bold lower case symbols for vectors, e.g., $\boldsymbol{a}$; (*iv*) bold upper case symbols for matrices and tensors, e.g., $\boldsymbol{A}$; (*v*) the position within a tensor or vector is denoted by numeric subscripts, for example, $\boldsymbol{A}_{ij}$ where $i, j \in \mathbb{N}^+$ indicates the element in the $i$-th row and $j$-th column of $\boldsymbol{A}$; (*vi*) calligraphic symbols for domains, e.g., $\mathcal{Q}$; (*vii*) time and step/layer dimensions are denoted as superscripts and everything else as a subscript, like the embedding for node $v_i$ at time $t$ and layer $k$ would be denoted as $z_i^{t,k}$.

## B  DATASET FEATURES

This appendix presents a more detailed explanation of the features considered in building the dataset. Specifically, these features include typical financial metrics — close price, open price, high, low, change, percent change, and volume-weighted average price (VWAP) — and technical indicators — Average Directional Index (ADX), Relative Strength Index (RSI), Williams %R, Double Exponential Moving Average (DEMA), Triple Exponential Moving Average (TEMA), Weighted Moving Average (WMA), and Standard Deviation. Specifically:

1. *Volume-Weighted Average Price (VWAP).* VWAP represents the average trading price of a security during a specific period adjusted so that trades with more volume have more influence, i.e., weighted by traded volume:

$$\text{VWAP} = \frac{\sum_{i=1}^{\tilde{n}} p_i v_i}{\sum_{i=1}^{N} v_i},$$

where $p_i$ is the price of the $i$-th trade, $v_i$ is its traded volume, and $\tilde{n}$ is the total number of trades.

2. *Average Directional Index (ADX).* In simple terms, ADX measures how strong a price trend is by the amount of upward and downward movements that dominate over a period of time, or mathematically, it quantifies the strength of a price trend over $\tilde{n}$ periods:

$$\text{ADX} = \frac{1}{\tilde{n}} \sum_{n=1}^{\tilde{n}} x_n,$$

$$x_n = \left| \frac{d^+ - d^-}{d^+ + d^-} \right| \times 100,$$

where $d^+$ and $d^-$ are the positive and negative directional indicators for period $n$.

3. *Relative Strength Index (RSI).* RSI measures recent price momentum to indicate overbought or oversold conditions:

$$\text{RSI} = 100 - \frac{100}{1 + r},$$

where $r$ is the ratio of average gains to average losses over a lookback period (commonly 14 days).

4. *Williams %R.* Williams %R shows where the current closing price is placed within the recent high-low range:

$$\%R = \frac{h - c}{h - l} \times -100,$$

where $h$ is the highest price, $l$ is the lowest price, and $c$ is the current closing price during the lookback period (typically 14 days).

5. *Double Exponential Moving Average (DEMA).* DEMA reduces lag compared to a simple EMA and is able to catch trends quicker:

$$\text{DEMA} = 2 \times \text{EMA}(p) - \text{EMA}(\text{EMA}(p)),$$

where $\text{EMA}(p)$ is the exponential moving average of the price series $p$ over $\tilde{n}$ periods.

6. *Triple Exponential Moving Average (TEMA).* TEMA further smooths data and reduces lag:

$$\text{TEMA} = 3 \times \text{EMA}_1 - 3 \times \text{EMA}_2 + \text{EMA}_3,$$

where $\text{EMA}_1 = \text{EMA}(p)$, $\text{EMA}_2 = \text{EMA}(\text{EMA}_1)$, and $\text{EMA}_3 = \text{EMA}(\text{EMA}_2)$.

7. *Weighted Moving Average (WMA).* WMA gives more weight to recent prices:

$$\text{WMA} = \frac{2}{n(n+1)} \sum_{i=1}^{n} p_i(n - i + 1),$$

where $p_i$ is the price at period $i$ and $n$ is the total number of periods.

8. *Standard Deviation ($\sigma$).* Standard deviation measures price volatility:

$$\sigma = \sqrt{\frac{1}{n} \sum_{i=1}^{n} (x_i - \mu)^2},$$

where $x_i$ is the price at time $i$, $\mu$ is the mean price, and $n$ is the number of observations.

## C  DATASET STATISTICS

This appendix presents the statistics regarding the dataset built in the proposed study. Specifically, Table 4 presents the main statistics from which it is visible that the temporal lead-lag graph consists of 1257 time steps, 575 less than the total number of days from June 12$^{\text{th}}$, 2019 to June 17$^{\text{th}}$, 2024 (1832 days), which is the result of omitting stock closure days. On average, each node has a degree of 2.99, with a standard deviation of 3.16. The average number of links per day is 14.04 with a standard deviation of 34.82, indicating considerable variability in the connectivity over time. This means there are periods with sparse and dense connections, as noticeable in Figure 3.

| Lead Node | Lag Node | Total Links |
|---|---|---|
| Lucid Group | SunRun | 76 |
| Natural Gas | SunRun | 73 |
| Solaredge Technologies | SunRun | 64 |
| Livent Corp | SunRun | 61 |
| SunRun | Solaredge Technologies | 59 |

Table 4: The five most connected pairs of nodes in the lead-lag temporal graph with $\epsilon = 5\%$ and $\tau = 1$ day.

| Statistic | Value |
|---|---|
| Number of time steps | 1257 |
| Number of nodes | 37 |
| Average node degree | 2.99 |
| Standard deviation node degree | 3.16 |
| Average number of links | 14.04 |
| Standard deviation number of links | 34.82 |

Table 5: Statistics of the lead-lag temporal graph with $\epsilon = 5\%$ and $\tau = 1$ day.

The average clustering coefficient, plotted in Figure 3, varies dynamically over time, typically oscillating around $0.5$. This metric provides insight into the local density of connections, and for a single node $v$ is given as:

$$c_u = \frac{2 \times \sqcup(v)}{\lceil(v)(\lceil(v) - 1)}$$

such that $\sqcup(v)$ denotes the number of triangles through node $v$ and $\lceil(v)$ is the degree of $v$. The final clustering coefficient for the whole graph is obtained as the average of the node coefficients. A clustering coefficient near $0$ typically indicates a disordered graph structure, whereas a value approaching $1$ suggests that nodes are likely to form densely connected clusters. Intermediate values imply that, on average, nodes possess a moderate degree of interconnected neighbours, neither highly isolated nor completely connected.

A significant metric for link prediction is the link probability (Manninen, 2022), shown in Figure 3, which refers to the proportion of new links formed at each time step relative to the total possible links in the graph. For a snapshot at time $t$, the link probability $p_t$ is defined as:

$$p_t = \frac{\text{New links at time } t}{n \times (n - 1)}$$

where $n$ is the total number of nodes in the graph, excluding self-loops. A low average link probability is observed, indicating that new links form infrequently. This is consistent with the sparsity of connections, as illustrated in the same figure by the number of connections per week. Upon further examination of Figure 3, a spike is observed at the end of the first quarter of 2020, likely due to increased price volatility during the COVID-19 pandemic breakout (Gormsen & Koijen, 2020).

With the complete definition of the lead-lag graph reported in the main paper, Table 5 highlights the five most frequently interconnected pairs of nodes, showcasing a distinct connection pattern among stocks in the solar energy sector, batteries, and electric vehicles. Caution is advised when drawing conclusions from this data, as the observed patterns may simply reflect the heightened volatility characteristic of these specific stocks.

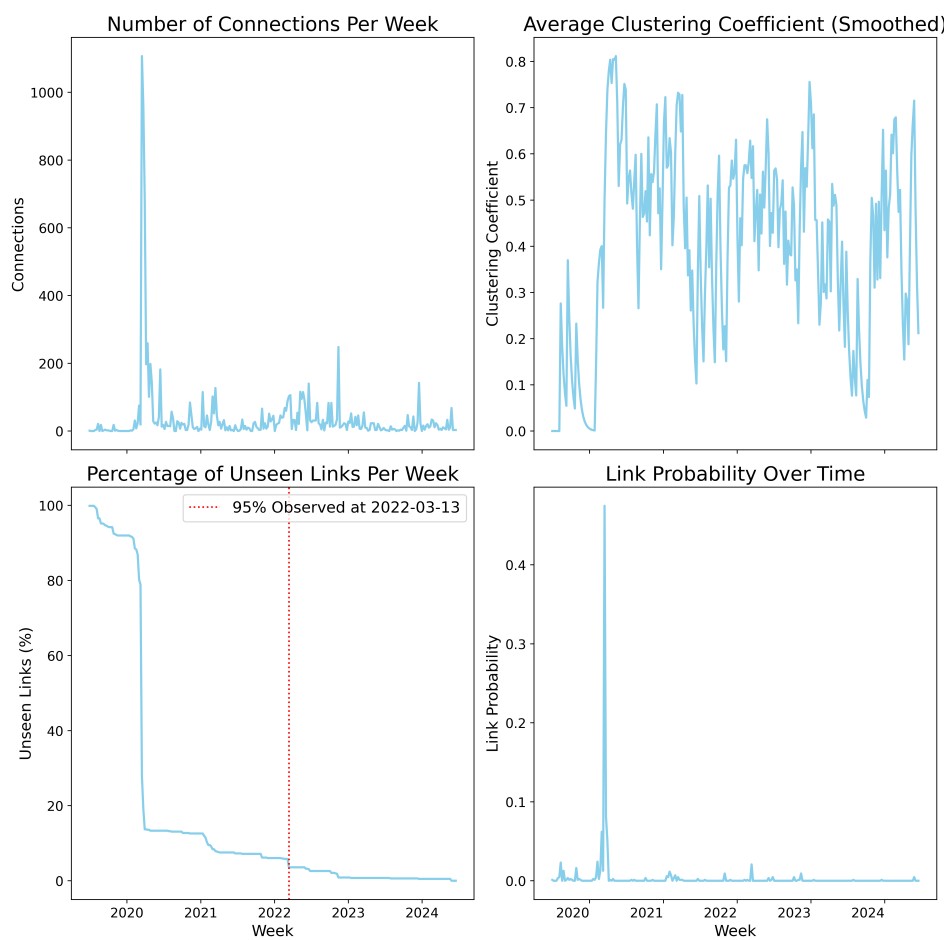

Figure 3: Temporal evolution of the lead-lag graph. Top: weekly connectivity with the number of connections (left) and smoothed clustering coefficient (right). Bottom: link behaviour with unseen links percentage (left, 95% observed marked) and link probability (right). Weekly aggregation is for presentation purposes only.

## D    EVALUATION METRICS

To objectively assess the models' performances, this work adopts several key metrics already adopted in the State-of-the-Art (Cong et al., 2023), which collectively provide a thorough evaluation of the model's ability to rank and identify relevant temporal links:

1. *Average Precision (AP)*. It measures the Precision-Recall trade-off by computing a weighted mean of Precision values at various Recall levels. It accurately captures the model's ability to rank relevant instances above irrelevant ones, crucial in imbalanced classification scenarios.

2. *Area Under the Curve (AAUC)*. It quantifies the area under the ROC curve.

3. *Recall at $k$ (R@k)*. It measures the proportion of relevant links retrieved among the top $k$ predictions. This metric evaluates the model's effectiveness in prioritizing true positives within the top $k$ results.

4. *Mean Reciprocal Rank (MRR)*. It evaluates ranking quality by examining the rank of the first relevant prediction for each query. Higher MRR indicates a better ranking of relevant links near the top.

When assessing a model's performance across training, validation, and testing phases, it is essential to compare a given positive link sample with one or more negative link samples. A positive sample

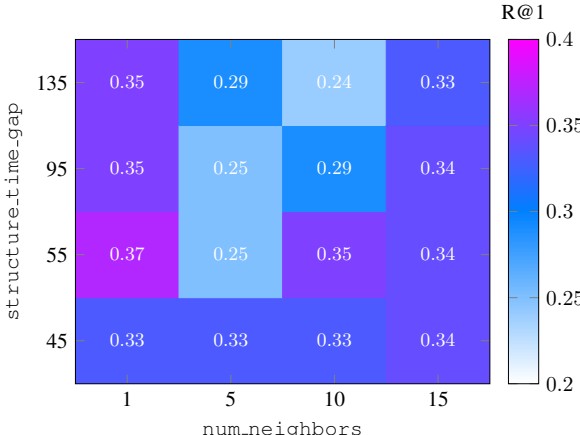

Figure 4: R@1 score for different combination of the `num_neighbors` and `structure_time_gap` hyperparameters for GraphMixer obtained after 20 epochs.

is precisely defined by the source node, the destination node, and the specific time step at which the link occurs. To calculate metrics like AP and AAUC, which involve comparing a positive sample to a single negative one, a node is randomly selected from the graph. This selection excludes the positive sample's source and destination nodes. The link formed between the positive sample's source node and this randomly chosen node then becomes the negative sample. Conversely, when evaluating the model's ranking capabilities, specifically metrics like R@$k$ and MRR, all potential links between the source node of the positive sample and any node, other than its actual destination node, are considered as negative samples. This comprehensive approach allows for a robust assessment of the model's ability to differentiate true positives from a wide range of negatives. It is important to acknowledge that this methodology may lead to the inadvertent selection of a true positive link (or multiple positive links, in the case of ranking evaluations) as a negative sample. However, this phenomenon can't be avoided since at a given evaluation time, say $t_0$, the graph lacks prior information regarding other links that may exist concurrently with the current positive sample being assessed, making the presented sampling strategy a practical and necessary compromise.

## E   MODEL SELECTION AND IMPLEMENTATION DETAILS

All models are trained using the training set and validated using the validation set. The model that performs best based on the validation set loss is chosen and then evaluated on the test set to obtain the final predictions. Specifically, the models are validated on the dataset considering both positive and negative lead-lag relationships, and then adopted "as-is" on the dataset made of only bullish trends. A grid search was conducted to identify the optimal hyperparameters. For each model, the best batch size was selected among the values in $[64, 128, 256]$, considering the AP metric. Specifically for GraphMixer, the hyperparameter tuning also focused on optimizing the `num_neighbours` and `structure_time_gap` parameters to enhance performance as measured by the R@1 score on the validation set. In particular, the value of `num_neighbours` is selected among $[1, 5, 10, 15]$ while the value of `structure_time_gap` is selected among $[45, 55, 95, 135]$.

Figure 4 presents varying performance levels obtained after training GraphMixer for 20 epochs with different combinations of these hyperparameters, using node features derived from embeddings obtained via a sentence transformer on company and commodity descriptions generated by an LLM. Notably, setting the `structure_time_gap` to 55 with `num_neighbours` set to 1 achieved the highest R@1 score, indicating that smaller structural time gaps and fewer neighbours may improve performance in certain settings. In contrast, configurations such as `structure_time_gap` = 55 and `num_neighbours` = 5 achieved lower scores, suggesting potential issues like over-smoothing or information dilution in more complex sub-graphs when too many neighbours are considered.

Thus, the selected best values are 128, 55, and 1, for the batch size, `structure_time_gap`, and `num_neighbors` parameters, respectively. Additionally, the model is trained with the binary cross-entropy loss function, with a learning rate of 0.0001 using the Adam optimizer, for 90 epochs.

The experiments were conducted using Python 3 on a Google Compute Engine back-end with GPU support, utilizing an NVIDIA T4 GPU on Google Colab. PyTorch 2.4.0 was employed as a back-bone deep learning framework, along with the associated libraries, torchvision 0.19.0 and torchaudio 2.4.0, installed via the PyTorch CUDA 12.1 package index. The PyTorch Geometric library was also employed for graph-based computations, with dependencies such as `torch-scatter`, `torch-sparse`, `torch-cluster`, and `torch-spline-conv` installed through the official PyG package repository. Moreover, `pybind11` was utilized to facilitate C++ integration for sampling operations.

The sentiment analysis used in building the dataset is performed using *company-name*[4] sentiment API, which calculates daily sentiment for each asset based on all accessible textual sources and assigns a confidence score based on the data available for the specific source on the given day.

## F    STATISTICAL TESTS

To rigorously assess the statistical significance of performance differences, this study conducted a comprehensive statistical hypothesis test. It first applied the Friedman test, which ranks model accuracies per dataset run and evaluates the null hypothesis that all models perform equivalently. Subsequently, it employed Conover's post-hoc test (Demsar, 2006), with the results visualized in the Critical Difference (CD) diagrams visible in Figure 2. The CD diagrams illustrate that the observed performance differences are statistically significant across the considered models. In both scenarios, GraphMixer (GM) achieves the best average rank, indicating a clear performance advantage. Its variant, GM-TNF, consistently ranks second, confirming that the temporal neighborhood aggregation contributes positively, although the improvement over the base GM model is moderate. JODIE and TGN occupy intermediate ranks, showing that they can capture temporal dynamics but are less competitive when compared to the simpler GM approach. Among the remaining models, DySAT and TGAT show limited performance, consistently ranking in the lower half. Finally, sequential architectures such as APAN and LSTM achieve the worst ranks in both diagrams. Their placement at the far right of the CD plots indicates that their differences with the top-ranked models are statistically significant, confirming that sequential models are insufficient to capture the structural and temporal dependencies of the lead–lag detection task. Notably, the difference between the two scenarios, considering both positive and negative relationships versus only positive relationships Figure 2), does not alter the overall ranking pattern. GM and GM-TNF remain statistically superior, while LSTM and APAN remain the weakest performers. This consistency reinforces the robustness of the conclusions drawn from the statistical analysis.

## G    ABLATION STUDY

Table 6 reports the performance of all the approaches using different kinds of link attributes:

1. *Description Embeddings*. Each node in the graph — i.e., a company or commodity — is represented with a 384-dimensional vector created by converting descriptions obtained from a language model into vector form. Link attributes are then composed by the embeddings of both source and destination nodes, i.e., a 768-dimensional vector.

2. *Embeddings + Prices*. These features are composed of both the description embeddings and the closing price at time $t$. It results in 385-dimensional features for nodes and 770-dimensional features for links.

3. *Embeddings + Prices + Financial Indicators + Sentiment*. These features incorporate precedents, along with financial indicators and sentiment scores, resulting in a 400-dimensional vector for nodes and an 800-dimensional vector for links.

The table reports the evaluation of all methods across the metrics described in Appendix D.

---

[4]Anonymized for double-blind review.

| Model | Link Attribute Type | Metrics | | | | | |
|---|---|---|---|---|---|---|---|
| | | AP ↑ | AAUC ↑ | R@1 ↑ | R@5 ↑ | R@10 ↑ | MRR ↑ |
| JODIE | Embeddings | **0.74** ± **0.03** | **0.77** ± **0.03** | **0.20** ± **0.03** | **0.58** ± **0.07** | **0.83** ± **0.06** | **0.30** ± **0.03** |
| | + Prices | 0.68 ± 0.01 | 0.71 ± 0.01 | 0.19 ± 0.01 | 0.56 ± 0.05 | 0.82 ± 0.03 | 0.29 ± 0.01 |
| | + Financial Indicators + Sentiment | 0.69 ± 0.03 | 0.73 ± 0.03 | 0.14 ± 0.03 | 0.44 ± 0.03 | 0.68 ± 0.02 | 0.24 ± 0.02 |
| DySAT | Embeddings | **0.73** ± **0.00** | **0.75** ± **0.02** | **0.25** ± **0.02** | **0.59** ± **0.02** | **0.77** ± **0.01** | **0.31** ± **0.00** |
| | + Prices | 0.72 ± 0.02 | 0.74 ± 0.01 | **0.25** ± **0.01** | 0.54 ± 0.04 | 0.74 ± 0.02 | **0.31** ± **0.00** |
| | + Financial Indicators + Sentiment | 0.66 ± 0.03 | 0.70 ± 0.03 | 0.22 ± 0.01 | 0.49 ± 0.02 | 0.69 ± 0.01 | 0.28 ± 0.01 |
| TGAT | Embeddings | 0.68 ± 0.02 | 0.72 ± 0.03 | 0.24 ± 0.01 | 0.52 ± 0.01 | 0.72 ± 0.01 | 0.30 ± 0.00 |
| | + Prices | **0.70** ± **0.03** | **0.73** ± **0.03** | **0.28** ± **0.00** | **0.57** ± **0.01** | **0.77** ± **0.00** | **0.33** ± **0.01** |
| | + Financial Indicators + Sentiment | 0.69 ± 0.03 | 0.72 ± 0.02 | 0.21 ± 0.02 | 0.48 ± 0.02 | 0.69 ± 0.04 | 0.26 ± 0.01 |
| TGN | Embeddings | **0.73** ± **0.02** | **0.75** ± **0.02** | **0.22** ± **0.02** | **0.58** ± **0.02** | **0.83** ± **0.02** | **0.31** ± **0.02** |
| | + Prices | 0.71 ± 0.02 | 0.73 ± 0.02 | 0.16 ± 0.03 | 0.48 ± 0.07 | 0.76 ± 0.07 | 0.27 ± 0.03 |
| | + Financial Indicators + Sentiment | 0.68 ± 0.02 | 0.69 ± 0.04 | 0.15 ± 0.02 | 0.40 ± 0.03 | 0.65 ± 0.02 | 0.24 ± 0.02 |
| APAN | Embeddings | **0.66** ± **0.05** | **0.70** ± **0.05** | **0.13** ± **0.01** | **0.43** ± **0.02** | **0.71** ± **0.02** | **0.24** ± **0.01** |
| | + Prices | 0.64 ± 0.05 | 0.66 ± 0.05 | 0.09 ± 0.03 | 0.34 ± 0.06 | 0.58 ± 0.02 | 0.19 ± 0.03 |
| | + Financial Indicators + Sentiment | 0.62 ± 0.02 | 0.66 ± 0.02 | 0.06 ± 0.01 | 0.21 ± 0.03 | 0.38 ± 0.04 | 0.14 ± 0.01 |
| GM-TNF | + Prices | **0.75** ± **0.01** | **0.82** ± **0.01** | 0.38 ± 0.01 | 0.79 ± 0.00 | 0.95 ± 0.00 | **0.46** ± **0.00** |
| | + Financial Indicators + Sentiment | 0.73 ± 0.02 | 0.81 ± 0.01 | **0.39** ± **0.02** | **0.81** ± **0.03** | **0.98** ± **0.02** | 0.46 ± 0.01 |
| GM | Embeddings | 0.78 ± 0.01 | 0.84 ± 0.02 | 0.39 ± 0.01 | 0.84 ± 0.01 | **0.99** ± **0.01** | 0.46 ± 0.01 |
| | + Prices | 0.77 ± 0.01 | 0.83 ± 0.01 | **0.42** ± **0.01** | **0.86** ± **0.02** | **0.99** ± **0.01** | **0.47** ± **0.01** |
| | + Financial Indicators + Sentiment | **0.79** ± **0.01** | **0.85** ± **0.01** | 0.41 ± 0.02 | **0.86** ± **0.03** | **0.99** ± **0.01** | **0.47** ± **0.02** |

Table 6: Ablation study showing the impact of different types of link attributes on model performance across various metrics. For each model, the best result per metric is highlighted in bold. Note that GM-TNF equals GM when both adopt only Embeddings.

## H GM VS GM-TNF

Figure 5 shows the metrics at varying link attribute types, for GM and GM-TNF. For GM the description embeddings are utilized as node features in all cases, while the time-varying attributes are used exclusively for the links. On the other hand, for GM-TNF, the embeddings are omitted, as both the node and link features have temporal properties. Based on the results shown in Figure 5, GM consistently outperforms its variant GM-TNF, implying that GM-TNF does not show any noticeable improvement over the base GraphMixer. This suggests that in this context, the additional temporal node features did not contribute meaningful extra information. The lack of improvement in GraphMixer-TNF highlights that the temporal nature of the attributes of the lead-lag links alone is enough to catch the dynamics in the link prediction task.

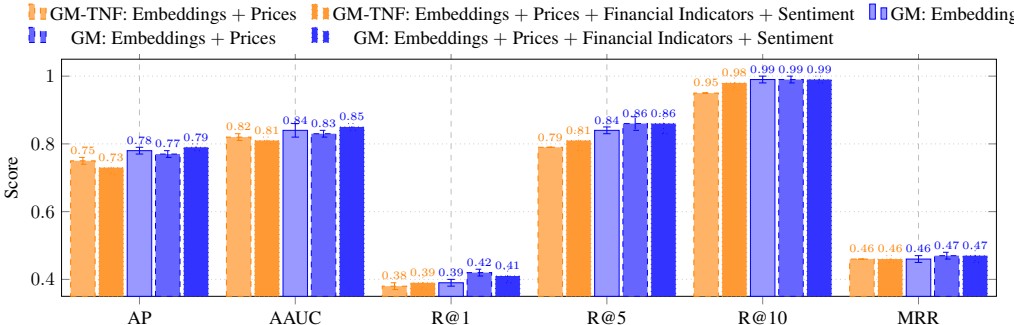

Figure 5: Barplot comparing GM and GM-TNF across metrics. Bar groups represent feature types — Embeddings, Embeddings + Prices, Embeddings + Prices + Financial Indicators + Sentiment — while the standard deviation is shown as error bar.

