# OpenReview forum: "A Temporal Graph Learning Framework for Lead-Lag Detection in Financial Markets"
_ICLR.cc/2026/Conference — ICLR 2026 Conference Withdrawn Submission_

### Official Review · Reviewer_G1wq · 2025-10-27

**Soundness:** 2
**Presentation:** 3
**Contribution:** 2
**Rating:** 4
**Confidence:** 4

**Summary:**

This paper addresses the widely studied lead–lag relationships in financial markets by framing the problem as a temporal link-prediction task and applying Temporal Graph Neural Networks (TGNNs). The authors construct a real-world benchmark of 37 assets spanning roughly five years of daily observations, generate training/test labels under two schemes (considering both up/down moves and positive-only moves), and adapt multiple models (sequence baselines like LSTM and several TGNN variants). Models are evaluated with binary classification and ranking metrics including AP, AAUC, MRR, Recall@k. Results show that TGNN approaches, in particular GraphMixer, achieve the most accurate and stable performance across most metrics, significantly outperforming the LSTM baseline.

**Strengths:**

- The paper addresses a common and highly significant problem in the financial domain of the discovery and detection of lead–lag relationships, aiming to fill the gap left by existing deep learning methods in tackling this issue.
- The proposed method demonstrates strong originality, and the experimental section effectively explores how different types of relationships and link attributes impact the model’s performance across various metrics.

**Weaknesses:**

- In the domain of financial time series, relationships between entities are highly transient, and at the daily frequency it is difficult to assert the existence of persistent multi-day lead–lag effects. In this paper, however, any short-term co-movement observed on a single day is labeled as a positive sample, which introduces substantial noise into the dataset. As the authors themselves note, their model attempts to capture “lead–lag relationships” rather than statistically validated “lead–lag effects,” meaning that many detected links may lack statistical or economic significance.
- The paper fixes the parameters at $\tau=1,\epsilon=5\%$​, a single arbitrary configuration that undermines the robustness of its findings. Furthermore, the construction of lead–lag links through a link-prediction task in a TGNN framework remains a complete black box. It is unclear why the authors did not cross-validate these learned relations against established econometric or sequence-comparison methods such as Granger causality tests, correlation analysis, or Dynamic Time Warping (DTW). Solely relying on ablation studies provides limited evidence for the real-world validity and consistency of the captured relationships.
- Lacks intuitive visualization or clear qualitative illustrations of the constructed lead–lag links. Without graphical representations such as temporal evolution graphs, network snapshots, or case studies makes it hard to assess whether the model captures economically meaningful dependencies or merely statistical artifacts.
- The work entirely omits any practical evaluation in real financial markets, such as price forecasting or portfolio backtesting across multiple assets. Without testing how the detected lead–lag relationships translate into predictive or trading performance, it is impossible to judge their actual utility or robustness under realistic market conditions.
- The choice of using 384-dimensional asset description embeddings as base inputs is questionable. As a static vector, it is unclear how this representation can capture time-varying lead–lag dynamics rather than merely encode long-horizon, cross-sectional priors. Moreover, such high-dimensional embeddings can easily dominate or distort lower-dimensional financial time-series signals, leading to representational imbalance; for sequential and graph-based models (e.g., LSTM, TGNN), this may hinder effective learning of temporal dependencies. Finally, the main text provides limited architectural details (with some hyperparameters deferred to the appendix), leaving a full layer-by-layer specification and dimensionalities insufficiently documented.
- The dataset contains only 37 assets sampled at daily frequency, which severely limits the generalizability of the results. This small-scale setting cannot represent broader markets with diverse sectors or liquidity levels, and it is inadequate for evaluating models on higher-frequency data where lead–lag effects are more likely to persist or recur.

**Questions:**

- How do the authors justify treating any single-day co-movement as a valid lead–lag instance rather than transient noise? Have they evaluated the model’s robustness under stricter labeling criteria, such as requiring multi-day persistence or repeated occurrences within a time window?
- Why did the authors not benchmark their detected lead–lag links against classical econometric or statistical approaches to validate that the TGNN-derived relations correspond to statistically meaningful effects?
- How does authors ensure that the predicted “lead–lag links” reflect genuine economic dependencies rather than spurious statistical correlations? Is there any interpretability mechanism (e.g., attention visualization or feature attribution) that can help explain what drives a predicted link?
- Since the paper motivates the study with potential trading and portfolio applications, why was no backtesting or forecasting evaluation performed? How do the authors envision using the detected lead–lag patterns in practical decision-making？
- Given the Efficient Market Hypothesis suggests that stock prices already fully reflect all available information, can the authors provide any theoretical justification showing that their method could achieve greater stability or consistency compared to traditional statistical or price-sequence-based approaches?

---

> ### Author Response · Authors · 2025-11-27
>
> We appreciate the reviewer's insights and are pleased that the problem addressed in our study is recognized as a highly significant problem in the financial domain. In addition, we are glad the reviewer recognized the originality of our proposed method, and the comprehensive exploration of how different relationship types and link attributes affect model performance. Below, we address the weaknesses highlighted by the reviewer.
>
> - **W1/Q1**: While it is true that we use $\tau=1$, which might initially suggest capturing also lead-lag relationships that aren't statistically significant in real-world scenarios, our results demonstrate that the models can predict these relationships over extended periods. This capability indicates that the models have learned meaningful patterns, which, by definition, cannot be attributed to noise and therefore represent real relations. Furthermore, traditional statistical methods in the current state of the art identify lead-lag effects by detecting persistent lead-lag relationships over a certain timeframe (i.e., 1 month as a period of time). However, this timeframe can vary significantly between different stocks, and the chosen time windows aren't always representative. Our approach, in contrast, can dynamically learn temporal patterns tailored to the specific stocks under consideration.
>
> - **W2/Q2**: We appreciate the reviewer's feedback. However, the values of $\tau = 1$ and $\epsilon$ were not chosen arbitrarily. As explained in Section 3.2, these values are based on established literature (Li et al., 2021; 2022; Sheth et al., 2023). Additionally, the dataset, sourced via an API from a company specializing in financial investments, recommended these values. Regarding other statistical approaches (as outlined in Section 3.1), our aim is not to compare our model against them, as they do not apply to the new problem formulation we propose. Traditional methods typically focus on pairwise or static analyses, whereas our approach allows for the simultaneous modelling of multiple assets with interdependent temporal dynamics. This facilitates capturing intricate, time-evolving lead-lag relationships across the market. To represent this complex data, we use a graph-structured data representation, where nodes represent assets and edges capture dynamic interactions. So, their adaptation to graphs is a non-trivial problem to address.
>
>  - **W3/Q3**: We appreciate the reviewer's suggestion. In the revised version of the paper, we intend to include visualizations of the predicted lead-lag interactions, emphasizing how they align with actual effects during specific time periods. These visualizations will offer an intuitive demonstration of the prediction quality and further highlight the practical significance of modeling temporal graphs to capture dynamic inter-asset relationships in financial markets.
>
> - **W4/Q4**: We targeted our submission to a computer science venue, which necessitated a focus on metrics pertinent to this field to ensure relevance and engagement with its audience. Including financial metrics would have shifted the emphasis away from the core interests of computer science researchers, potentially diminishing their impact and relevance. However, we acknowledge the rich potential for exploring these financial aspects more deeply. This presents an opportunity for a future submission to a financial venue, where we can delve into those dimensions more thoroughly, offering substantial material for another paper.
>
> - **W5**: The embedding evolves over time, allowing the model to capture the dynamics of the phenomena through the evolution of 384 feature values. Therefore, the model's input is not a simple vector but a sequence of data points. Additionally, the model incorporates lower-dimensional financial time-series signals into its input. The results suggest that the model effectively filters out irrelevant information while identifying lead-lag relationships, even between distant points in the series. Regarding the model's hyperparameters and architecture, they were determined through a process of model selection, detailed in the main article, Section 4.2 and Appendix E, to ensure the results are reproducible. Although the specific hyperparameter values are not included in the main paper, as we consider them less critical than the selection methodology, we provide access to the complete code submitted as Supplementary Material.

---

> > ### Author Response · Authors · 2025-11-27
> >
> > - **W6**: While the dataset includes a moderate number of assets, it is derived from real financial market data within a specific sector and spans a period marked by major disruptions, including the COVID-19 crisis. Such events introduce significant temporal shocks and shifts in lead–lag dependencies, making the temporal modeling task highly challenging despite the graph’s size. Additionally, because the assets belong to a closely connected sector, the resulting network exhibits dense and complex interactions, further increasing the difficulty of modeling dynamic relationships over time. Thus, even with a relatively limited number of nodes, the dataset serves as a meaningful and demanding benchmark for evaluating the proposed TGNN framework’s ability to capture intricate, evolving dependencies.
> >
> > - **Q5**: Unfortunately, it is not feasible, because in our proposed approach, we modelled the problem as a temporal graph instead of the traditional price sequences, where statistical tests are usually conducted between pairs of stocks. Our goal is to explore and understand complex economic lead-lag patterns by considering multiple assets simultaneously (through the graph). This involves a novel way of framing the problem that offers perspectives beyond conventional statistical methods. Consequently, making a theoretical comparison is more challenging since the two modelling approaches are fundamentally different, and applying standard statistical methods to graphs is not straightforward (Section 3.1). Adaptations would require developing new statistical techniques that significantly diverge from the original definitions.

---

### Official Review · Reviewer_ZxC3 · 2025-10-28

**Soundness:** 2
**Presentation:** 1
**Contribution:** 3
**Rating:** 2
**Confidence:** 4

**Summary:**

This paper presents a novel framework for detecting financial lead-lag relationships. Its core contribution is to reframe this classic statistical problem as a temporal link prediction task on a dynamic graph. By doing so, it moves the analysis from isolated, pairwise comparisons to a holistic, network-level problem, allowing the application of modern Temporal Graph Neural Networks (TGNNs). The authors provide a new benchmark dataset and a rigorous comparison of seven TGNNs, ultimately finding that a simple, MLP-based model (GraphMixer) outperforms more complex architectures.

**Strengths:**

- Transforming lead–lag detection into temporal graph link prediction is a generalizable framing that moves the topic from econometrics to machine learning.
- The dataset construction process (returns, indicators, LLM embeddings, sentiment) is described in fair detail, which encourages reproducibility.
- The use of Friedman and Conover's post-hoc tests (Figure 2) elevates the results from a simple "league table" to a statistically validated conclusion.

**Weaknesses:**

- The entire framework rests on two "magic numbers," $\tau=1$ day and $\epsilon=5\%$, which are never justified through a sensitivity analysis. The finding that GM is the best model is contingent on this specific, restrictive definition of a "lead-lag," and the results may not generalize to different time lags or thresholds.
- The paper claims to model both short-term "relationships" and long-term "effects" but then uses a fixed $\tau=1$ day lag for all graph construction. By definition, this architecture cannot capture longer-term (e.g., weekly or monthly) effects, making its initial framing misleading.
- The graph contains only 37 nodes, which is a very small network. The performance and ranking of these models could change dramatically on a larger, more realistic graph, where neighborhood aggregation and scalability become non-trivial problems.
- The introduction spends too much space on dataset and experimental description instead of building conceptual motivation or framing research gaps.
- Some strong, absolute claims are immediately contradicted by the authors themselves in the very next sentences, where they refine the claim to "existing studies rarely leverage graph-based representations, and when they do, they typically consider static rather than dynamic structures".

**Questions:**

Please see weakness above.

---

> ### Author Response · Authors · 2025-11-27
>
> We appreciate the reviewer's effort in highlighting key points and are pleased that our novel framing of lead-lag detection as temporal graph link prediction, our commitment to reproducibility through detailed dataset construction, and our use of statistical tests to strengthen our findings were recognized. In the following, we address each weakness highlighted by the reviewer.
>
> - **W1**: We appreciate the reviewer's feedback. However, the values of $\tau=1$ and $\epsilon$ are not chosen arbitrarily. As detailed in Section 3.2, these values are selected based on established literature (Li et al., 2021; 2022; Sheth et al., 2023). Additionally, as mentioned in Section 3.2, the dataset is sourced via an API from a company specializing in financial investments, which also recommended these values.
>
> - **W2**: In our proposed approach, we set $\tau=1$, which corresponds to a single day, to capture high-frequency dynamics. By modeling the problem as a temporal graph, where assets at time $t$ can be linked to assets at time $t+1$, graph neural networks can learn lead-lag relationships across multiple days. This ability is a key advantage of using temporal graph link prediction for our problem. Note that the relationships are detected considering all the co-occurrences of lead-lag patterns among the assets throughout time, not just considering pairs of them.
>
> - **W3**: While the graph contains a moderate number of nodes, it is constructed from real financial market data and is representative of the sector under investigation, covering a period that includes major market disruptions such as the COVID-19 crisis. These events introduce strong temporal spikes and shifts in the lead–lag dependency structure, making the temporal modeling problem highly non-trivial despite the graph’s size. Moreover, since the assets belong to the connected sector, the resulting network exhibits a dense and complex interaction structure, increasing the challenge of capturing dynamic relationships over time. Therefore, even with a relatively small number of nodes, the dataset provides a meaningful and demanding testbed for evaluating the proposed TGNN framework, especially in terms of its ability to model intricate, time-evolving dependencies. It is also important to note that the scalability and efficiency of GNN-based models on large-scale graphs have already been widely studied in the literature, confirming their applicability to larger datasets in terms of both performance and computational feasibility. In particular, the Graph Mixer architecture is a relatively simple yet effective model, which offers favorable computational efficiency and makes it well-suited for scaling to larger and more complex graphs.
>
> - **W4**: We respectfully disagree with the reviewer, as explaining how the dataset is gathered is a critical decision point for us, particularly for ensuring reproducibility. This is especially important as the features selected to build the dataset significantly affect the results. Additionally, we do not concur with the reviewer’s view because we have also devoted considerable space in Section 1, Section 2, and Section 3 to outline the research gap that this work addresses.
>
> - **W5**: Could you kindly indicate any specific absolute claims we have made? We would appreciate any references to prior works addressing the same problem in the same manner. Additionally, could you please clarify what you mean by contradiction? In the sentence you provided, we do not perceive any contradiction with what we have presented in the article.

---

### Official Review · Reviewer_cYy7 · 2025-10-30

**Soundness:** 2
**Presentation:** 3
**Contribution:** 2
**Rating:** 2
**Confidence:** 3

**Summary:**

This paper focuses on the importance of the lead-lag relationship in financial markets, which is proposed to be expressed in a temporal graph structure, with the detection of their effects as a temporal link prediction task. Based on this reformulation, the authors introduce a new benchmark for the performance comparison of SOTA temporal GNNs on modeling lead-lag interactions that can be both positive and negative. Experiments are evaluated on a custom dataset built upon financial assets based on temporal, structural, and sentiment features.

**Strengths:**

The main strengths of this work are summarized as follows:
1. The problem of inter-stock interactions (often expressed as a lead-lag effect) forms indeed an interesting modeling aspect in applications on financial markets; therefore, experimental comparisons on predicting the strength of lead-lag interactions can be useful (Significance).
2. Stock price prediction depends on external information; therefore, evaluating on a dataset built on interactions and sentiment features beyond pure price movements highlights the need for adding external knowledge in this domain (Quality).
3. The benchmark is easy to follow, and results include statistical significance tests (Clarity).

**Weaknesses:**

The weak aspects of the paper are the following:
- *(W1)* The authors disregard mentioning related works on hypergraphs for stock prediction tasks [3], some of which explicitely consider lead-lag relationships [1,4,5]. The related work should be enhanced such that the most recent works leveraging GNNs and graph interaction for stock data is presented. Additional recent models in stock prediction could be mentioned [2].
- *(W2)* It is unclear why predicting lead-lag interactions alone is so crucial, while in the relevant literature the end task is most of the time stock price prediction. The authors should justify their choice or experimentally showcase how the prediction of interaction can improve forecasting performance.
- *(W3)* For people aware of financial markets, it is common knowledge that stocks belonging in the same sector interact positively (and based on their size can lead smaller stocks), while stocks in competing sectors interact negatively. It is unclear how the proposed custom-made dataset captures complex interaction patterns, which is crucial since the final task is indeed predicting the interactions. The authors should mention also why existing datasets (e.g., those used in related works) are not sufficient for lead-lag modeling.
- *(W4)* Based on W3, to showcase how “easy” the constructed dataset is for lead-lag identification, including if temporal graphs are better than static, simpler models that build on correlations should be considered, e.g., graph-based baselines on similarity/causality measures (pairwise Granger-causality). Different experimental setups are crucial to prove the necessity of predicting dynamic interactions and could potentially be combined with an end task (e.g., forecasting).
- *(W5)* Including some visualizations on the predicted lead-lag correlations and how this is indeed connected to true effects during specific periods based on temporal patterns, could be a complementary way to prove the necessity of temporal graphs in the field. In the current form of the paper it is unlear how predicting the power of an interaction can be used in practice for solving tasks in financial markets.
- *(W6)* The paper proposes a benchmark, yet fails to position itself against other benchmarks in the field (if existent) and why those are not enough for tasks in financial markets. The methododological novelty is limited since the authors do not propose a method but rather a whole family of methods that should be assessed, while the construction of the dataset seems a bit heuristic (not justified as an extension of prior works).

[1] HUYNH, Thanh Trung, et al. Efficient integration of multi-order dynamics and internal dynamics in stock movement prediction. In: Proceedings of the Sixteenth ACM International Conference on Web Search and Data Mining. 2023. p. 850-858.

[2] FAN, Jinyong; SHEN, Yanyan. StockMixer: A simple yet strong MLP-based architecture for stock price forecasting. In: Proceedings of the AAAI conference on artificial intelligence. 2024. p. 8389-8397.

[3] LIAO, Sihao, et al. Stock trend prediction based on dynamic hypergraph spatio-temporal network. Applied Soft Computing, 2024, 154: 111329.

[4] LI, Yongli, et al. Dynamic patterns of daily lead-lag networks in stock markets. Quantitative Finance, 2021, 21.12: 2055-2068.

**Questions:**

- *(Q1)* Based on (W1) and (W2), the authors should clarify the importance of a lead-lag detection benchmark for tasks involving financial market data, while properly positioning them against any work on machine learning (graph-based) methods in the field.
- *(Q2)* Based on (W3) and (W6), the authors should showcase why in practice existent datasets/benchmarks are insufficient and how the proposed dataset/benchmark can be leveraged for new studies in the field.
- *(Q3)* It is unclear whether the detection of lead-lag relationships is easy based on the extracted relationships and the link prediction task, and how temporal learning can be informative for stock markets, as expressed in (W4) and (W5).

---

> ### Author Response · Authors · 2025-11-27
>
> We appreciate the reviewer's feedback on our paper. We are pleased that our work in modeling lead-lag interactions for financial markets was recognized, and that the interactions and sentiment features needed were highlighted. We also appreciate the acknowledgment of our benchmark's clarity and the use of statistical significance tests. Below, we address the reviewer’s concerns.
>
> - **W1**: We thank the reviewer for highlighting these relevant works. We will update the related work section to include the mentioned hypergraph-based approaches for stock prediction and explicitly discuss how some of them consider lead–lag relationships. Additionally, we would like to highlight the significant differences between our graph-based approach and these hypergraph models, since the formulation of the problem considered in the reported references is significantly different from the one we considered. Hypergraph methods [1,3] model higher-order relationships through hyperedges connecting multiple stocks simultaneously, whereas our approach focuses on lead-lag between two assets/nodes relationships captured through directed graphs. Regarding [2], while StockMixer presents an interesting MLP-based architecture for stock prediction, it does not explicitly address lead-lag relationships. Additionally, we would like to point out that [4] is already cited and discussed in Section 2 "Background And Related Works" of the paper.
>
> - **W2**: We would like to clarify that our focus on predicting lead–lag interactions is motivated by the desire to model complex, interdependent temporal dynamics across multiple assets, rather than directly forecasting individual stock prices. As highlighted in Section 1 (Introduction), traditional methods often focus on pairwise or static analyses, which are insufficient to capture the evolving relationships that propagate through the market. By framing the problem as a temporal link prediction task on a dynamic graph, where nodes represent assets and edges capture their time-varying interactions, we can model these intricate lead–lag relationships efficiently. Predicting such interactions is a foundational step that can enhance downstream tasks, including price forecasting, by providing richer information on inter-asset dependencies. The use of TGNNs allows us to leverage established graph-based techniques while adapting them to this novel formulation, demonstrating the utility of capturing temporal interactions explicitly, which goes beyond traditional forecasting approaches.
>
> - **W3**: We would like to highlight that, since the dataset consists of stocks from similar sectors, the resulting graph is denser and more interconnected, which increases the difficulty of modelling. Moreover, the highly dynamic changes in connections make the temporal component critical to learn, thereby increasing the complexity of capturing both topological and temporal dependencies. This setting underscores the importance and relevance of our temporal graph-based approach.
>
> - **W4**: We agree with the reviewer that conducting a comparison with traditional econometric baselines, such as Granger causality, could further strengthen the rationale of our work. However, as discussed in Section 3.1 of the paper, such a comparison is non-trivial. Our approach frames lead–lag prediction as a link prediction task on a temporal graph, which fundamentally differs from traditional statistical methods. Adapting econometric baselines to this setting would require the development of hybrid approaches that, while inspired by the original methods, differ from them in key aspects. This divergence makes direct comparisons complex and not straightforward, but we acknowledge the potential value of exploring such adaptations in future work.
>
> - **W5**: We thank the reviewer for this suggestion. In the updated version of the paper, we plan to include visualizations of the predicted lead–lag interactions, highlighting how they correspond to true effects during specific periods. These visualizations will provide an intuitive demonstration of the quality of the predictions and further illustrate the practical relevance of modelling temporal graphs for capturing dynamic inter-asset relationships in financial markets.

---

> > ### Author Response · Authors · 2025-11-27
> >
> > - **W6**: We thank the reviewer for this observation. We would like to clarify that our work provides both a novel benchmark dataset and a problem formulation for lead–lag prediction on dynamic graphs. While other benchmarks may exist in financial data research, they generally focus on price forecasting or static correlations and do not capture multi-asset temporal interactions in the structured graph form that our benchmark provides. We will update the paper to explicitly position our benchmark relative to existing datasets and clarify its unique contribution for tasks in financial markets. Regarding methodological novelty, we emphasize that the work introduces a new task formulation for temporal GNNs, enabling the evaluation of a family of graph-based methods on this benchmark. We would like to clarify that the dataset construction is not heuristic. It is based on real-world data collected over multiple years, spanning key periods such as the COVID-19 crisis, and includes prices, financial indicators, and sentiment features for stocks and commodities. The dataset is therefore in representative of actual market dynamics and provides a meaningful benchmark for evaluating temporal graph-based models in realistic financial scenarios. We will update the paper to make this point explicit and highlight its grounding in empirical data rather than heuristic choices.
> >
> > Follow the answers to the questions of the reviewer:
> >
> > - **Q1**: In addition to our responses to W1 and W2, we would like to emphasize that lead–lag detection serves purposes that go far beyond price prediction. Establishing a benchmark for this task is therefore crucial for both research and practical applications in financial markets. Lead–lag relationships can inform risk management and early warning systems by identifying assets likely to propagate shocks, support pairs trading and statistical arbitrage strategies, guide portfolio rebalancing decisions, enable systemic risk monitoring for regulators, and provide insights into market microstructure dynamics. We will update the paper to position our benchmark relative to existing machine learning and graph-based methods and to clearly highlight these practical and research applications, demonstrating the broader relevance of lead–lag detection in financial markets.
> >
> > - **Q2**: In addition to our responses to W3 and W6, we would like to clarify that our intention is not to claim that existing datasets or benchmarks are insufficient. Rather, we present new data covering the last five years, which are both costly and time-consuming to collect. This dataset provides a richer and more comprehensive set of features than existing resources, including not only financial prices but also technical indicators, sentiment, and semantic embeddings. Such completeness enables models to learn from multiple types of signals, supporting research well beyond lead–lag detection. Consequently, the proposed benchmark can facilitate new studies in financial markets that leverage the combination of price, indicator, and sentiment information, offering broader opportunities than prior datasets.
> >
> > - **Q3**: In addition to our responses related to W4 and W5, we would like to emphasize that the detection of lead–lag relationships in our dataset is non-trivial. If lead–lag patterns were simple, such as sector co-movement, basic correlation-based methods would achieve near-perfect performance, which is clearly not the case. As shown in Figure 3, over 95\% of unique links appear only after the initial training period, the average link probability is below 5\% (indicating sparse connections), and there is high temporal variability in the interaction patterns. These characteristics require the model to generalize to novel relationships rather than memorize static structures. Moreover, both lead–lag patterns and stock prices are inherently temporal, and our work focuses on modeling how lead–lag relationships evolve over time, when they occur, and when they break, which is often challenging for traditional statistical methods. The temporal learning component is therefore crucial, as it allows the model to capture dynamic interactions that provide insight into market behavior, going beyond the simple task of predicting price movements.

---

### Official Review · Reviewer_TDwf · 2025-11-02

**Soundness:** 2
**Presentation:** 3
**Contribution:** 2
**Rating:** 4
**Confidence:** 4

**Summary:**

This paper investigates the problem of detecting lead–lag relationships among financial assets — that is, identifying which assets tend to move ahead of others in time. The authors argue that such temporal dependencies can be naturally represented as dynamic graphs, where nodes correspond to assets and edges indicate evolving lead–lag effects. To this end, they formulate the problem as a temporal link prediction task and evaluate several deep learning models.

A custom dataset of 37 financial assets over 1,257 time steps is constructed, incorporating price, technical indicators, and sentiment features. Experiments are conducted under two settings — one considering both positive and negative lead–lag relationships, and another focusing on positive (bullish) ones only. Results show that GraphMixer achieves the best overall performance across several ranking-based metrics (AP, AAUC, MRR, R@k), suggesting that temporal graph learning can effectively capture complex dependencies between asset movements. An ablation study and statistical significance analysis (via Friedman and Conover tests) are provided to support the empirical findings.

**Strengths:**

The paper is generally well written and easy to follow. The overall structure is clear. The proposed framework for modeling lead–lag relationships through temporal link prediction on dynamic graphs is conceptually intuitive and technically straightforward to understand. The authors also make an effort to compare a range of baseline models, from simple sequence models (e.g., LSTM) to state-of-the-art temporal GNNs, which helps situate their approach within the broader literature.

**Weaknesses:**

1) The motivation of the current paper is not sufficiently solid. Lead–lag analysis has a well-established foundation in financial econometrics, and the paper fails to clearly articulate the specific limitations of traditional approaches—such as Granger causality, transfer entropy, or temporal causal graphs—in modeling dynamic dependencies. Moreover, it does not demonstrate any distinctive advantage of TGNNs in capturing causal directionality or improving predictive interpretability. Although the proposed models show some empirical performance gains, the work lacks a theoretically novel problem formulation, making its motivation appear overly generic.

2) From a methodological perspective, the proposed approach lacks theoretical justification, particularly given that financial markets are high-dimensional stochastic systems with strong non-stationarity, heavy-tailed noise, and structural regime shifts. The paper does not provide any analysis—either theoretical or empirical—regarding the reliability and robustness of using temporal graph neural networks (TGNNs) for modeling such inherently unstable dynamics. While the model achieves performance improvements on the constructed dataset, it remains unclear why the architecture should generalize beyond this specific sample or how it avoids overfitting to transient correlations. In the absence of a theoretical grounding (e.g., under what assumptions the TGNN can approximate Granger causality or causal directionality), the results risk being purely correlational and non-informative from an economic standpoint.

3) A major limitation of this paper lies in the scale and representativeness of the dataset. The dynamic lead–lag graph includes only 37 nodes (assets) and 1257 time steps, which is extremely small compared to real-world financial markets that typically involve hundreds or thousands of correlated instruments. Such a limited sample size effectively makes this a proof-of-concept experiment, rather than a validation of the method’s robustness or generalizability. Given the stochastic and non-stationary nature of financial systems, conclusions drawn from such a small and domain-specific subset (mostly renewable energy and EV-related stocks) cannot be confidently generalized to broader market settings. The paper should either (i) justify why this small-scale dataset is sufficient to demonstrate the claimed methodological advantages, or (ii) extend the evaluation to a larger, more diverse asset universe to substantiate its claims.

4) Another important omission concerns the model architecture specifications.
For all temporal GNN baselines (GraphMixer, TGN, DySAT, TGAT, JODIE, and APAN), the paper does not report fundamental architectural details such as hidden dimensionality, number of layers, activation functions, dropout rates, or message-passing depth.
These parameters critically influence both model capacity and generalization, and without them, it is impossible to assess whether the reported performance differences arise from methodological advantages or from arbitrary architectural tuning.

5) The paper tends to overclaim its novelty.
Essentially, it models lead–lag relationships in financial event sequences from a purely data-driven perspective, yet such relationships have been extensively studied in financial econometrics.
Many assets (e.g., derivatives such as futures and options) are structurally designed to exhibit lead–lag behavior with respect to their underlying assets.
Therefore, the problem setting itself is not new. If the contribution is mainly algorithmic, the paper should clearly position itself as a temporal graph learning approach for financial data.
Conversely, if it aims to make a domain contribution, it should provide a stronger economic rationale for why existing causal or econometric tools (e.g., Granger causality, transfer entropy, or high-frequency microstructure analysis) are insufficient.
Without such grounding, the paper reads as a technical application rather than a conceptually motivated advancement, and the claimed practical significance feels overstated.

**Questions:**

1) Could the authors explicitly articulate what concrete limitations of traditional econometric or causal methods (e.g., Granger causality, transfer entropy, temporal causal graphs) their TGNN approach overcomes? What aspect of “dynamic dependency modeling” cannot be addressed by these existing frameworks?

2) Under what theoretical assumptions can TGNNs be expected to capture causal directionality or Granger-like dependencies?
Have the authors considered any formal connection between message-passing dynamics and causal inference principles in stochastic systems?

3) Since the dataset contains only 37 assets and 1257 time steps, what justifies the claim that the approach generalizes to broader market settings?
Can the authors comment on how the model’s complexity scales with larger, more realistic asset universes?

4) The dataset appears to be dominated by renewable energy and EV-related stocks.
How might this domain bias affect the interpretability of the learned lead–lag relationships?
Are the results robust across other industry sectors or asset classes?

5) Beyond numerical performance gains, what do the discovered lead–lag relationships imply from an economic or behavioral perspective?
Can the authors show any concrete examples (e.g., lead–lag between futures and underlying equities) that correspond to known market mechanisms?

---

> ### Author Response · Authors · 2025-11-27
>
> We appreciate the reviewer's feedback on our paper. We are thrilled that the reviewer found our article clear and well-structured, understood our innovative approach in modeling lead-lag relationships through temporal link prediction on dynamic graphs, and valued the comprehensive nature of our experimental overview. Below, we address the concerns raised by the reviewer.
> - **W1/Q1**: We would like to clarify that our aim is not merely to highlight limitations of traditional statistical approaches, such as Granger causality, but rather to propose a novel problem formulation, as underscored in Section 1 (Introduction). Traditional methods often focus on pairwise or static analyses, whereas our approach enables the simultaneous modelling of multiple assets with interdependent temporal dynamics, facilitating the capture of intricate, time-evolving lead–lag relationships across the market. Then, to represent this complex data, we rely on a graph-structured data representation, where nodes represent assets and edges capture dynamic interactions. The use of TGNNs is therefore a natural choice, as they allow us to leverage established (and well-studied) graph-based approaches while adapting them to our lead–lag formulation. This combination highlights the novelty of our approach, both in terms of problem formulation and in showing the ability of TGNNs to model complex real-world temporal dependencies.
> - **W2/Q2**: We acknowledge that deep learning models can initially lack explainability. However, their strong performance metrics show they effectively predict lead-lag relationships, providing valuable economic insights retrospectively. Additionally, our objective is not to demonstrate how our model compares against the ``Granger Causality'' statistical test, as it does not apply to our problem formulation. Instead, we aim to explore and understand complex economic lead-lag patterns, which consider several assets simultaneously, using a novel approach to framing the problem, offering different perspectives beyond traditional statistical methodologies.
>  - **W3/Q3**: We acknowledge that the dataset is relatively small in terms of the number of nodes; however, we emphasize that it is based on real market data that are representative of the sectoral trends under study, covering a period that includes multiple significant market events, including the COVID-19 crisis. These events introduce pronounced temporal spikes and shifts in the lead–lag structure.  Furthermore,  the assets belong to a similar sector, and this increases the density and complexity of relationships, making the modelling of temporal dynamics more challenging. Thus, even with a moderate number of nodes, the dataset provides a meaningful and non-trivial testbed for assessing the proposed TGNN methodology and its ability to capture intricate, time-evolving relationships.
> - **W4/Q4**: In the article, specifically Section 4.2 and Appendix E, we have reported several implementation details.  Note that all the solutions we tested are directly derived from model architectures previously proposed in the literature (and cited in the paper). Due to space constraints, we provide a detailed description of how we adapted these architectures to the considered problem in Section 3.4. Moreover, we would like to clarify that we have made the code available through the submission, which includes comprehensive details about the modifications made to adapt each model to the proposed problem.
> - **W5/Q5**: We thank the reviewer for this detailed feedback. We would like to clarify that, contrary to the reviewer’s statement, our work proposes a new formulation of the lead–lag detection problem, framing it as a temporal link prediction task on dynamic graphs. This formulation differs fundamentally from traditional econometric or causal approaches, as it enables the simultaneous modelling of multiple interdependent assets over time and explicitly captures evolving graph structures. If there are existing works we may have overlooked, we would greatly appreciate any references the reviewer could suggest. Our paper aims to contribute not only algorithmically but also conceptually, by introducing a novel task for temporal GNNs in financial domains. To support this, we provide a new benchmark dataset covering stocks and commodities over five years, including daily prices, financial indicators, and sentiment features. This dataset allows the evaluation of TGNNs in a structured, multi-asset, temporal context, highlighting both the methodological and empirical contributions of our work.

---

### Official Review · Reviewer_Nq3D · 2025-11-02

**Soundness:** 3
**Presentation:** 2
**Contribution:** 2
**Rating:** 4
**Confidence:** 3

**Summary:**

This paper reformulates the classical lead–lag detection problem in financial markets as a temporal link prediction task on dynamic graphs. By representing assets as nodes and directional edges as lead–lag interactions, the authors aim to capture time-evolving interdependencies beyond traditional pairwise statistical methods such as Granger causality. The study builds a benchmark dataset of 37 entities (29 stocks and 8 commodities) covering five years of daily data enriched with financial indicators and sentiment features. It evaluates eight models — from LSTM baselines to several Temporal GNN (TGNN) architectures (JODIE, DySAT, TGAT, TGN, APAN, GraphMixer, and GraphMixer-TNF). Experiments compare performance across two settings: (1) both positive and negative lead–lag relationships, and (2) positive-only (bullish) relationships. The results show that GraphMixer, a simple MLP-based architecture, consistently outperforms more complex TGNNs, suggesting that model simplicity can suffice for this task.

**Strengths:**

1. Novel framing: The paper provides a fresh machine-learning-based formulation of lead–lag detection as temporal graph learning, moving beyond static or pairwise correlation-based approaches. This reconceptualization opens opportunities for applying TGNNs to dynamic financial dependencies.

2. Comparison fairness: The authors implement a fair experimental setting, using consistent frameworks (TGL) across models and appropriate evaluation metrics (AAUC, MRR, Recall@k), mitigating class imbalance concerns.

3. Empirical comprehensiveness: Comparative analyses across multiple architectures and ablation studies (feature type variations, statistical tests) provide a broad experimental overview.

**Weaknesses:**

1. Temporal heterogeneity and regime shifts overlooked: The dataset includes the COVID-19 crisis, during which the paper itself notes a “spike” in link formation density. Yet, the model evaluation does not examine robustness across different market regimes or stress periods. Without regime-based or rolling-window validation, it is unclear whether the model truly generalizes beyond specific volatility clusters.

2. Static features dominate performance: In the ablation study (Table 3), models using only static description embeddings achieve comparable or even better results than those incorporating temporal features such as price and sentiment. This implies that temporal dynamics — the central motivation of the paper — are not the main performance drivers. The dataset may therefore encode static inter-asset similarity rather than dynamic lead–lag propagation.

3. Lack of economic interpretability and validation: Despite references to “investment insights,” the study does not include trading simulations or backtesting to establish economic significance. Nor does it benchmark against traditional econometric baselines like Granger causality, VAR models, or rolling correlation networks, leaving its claims of superiority over “statistical methods” empirically unverified.

**Questions:**

1. Why do you think the model achieves nearly top-level performance even when using only static description embeddings?

2. Does the fact that a simple model like GraphMixer performs so well indicate that the temporal structure is not being properly captured — or perhaps that it is not as important as expected?

---

> ### Author Response · Authors · 2025-11-27
>
> We appreciate the reviewer's insightful comments and suggestions. We are pleased that the reviewer recognizes the innovative approach of framing lead–lag detection as a temporal graph using machine-learning techniques, acknowledges the fairness of our evaluation setting, and values the broad experimental overview. Below, we provide our responses to the weakness raised by the reviewer:
>
> -  **W1**: We thank the reviewer for this valuable observation and fully agree with the concern regarding temporal heterogeneity and regime shifts. In response, we will extend our evaluation by exploring training scenarios that go beyond the COVID-19 period. Specifically, we will exclude COVID-19 data from the training set and instead use it solely for testing. This will allow us to assess our model's ability to generalize to unseen stress conditions and better highlight its robustness across different market regimes, including periods of heightened volatility. We believe these enhancements will significantly strengthen the empirical validation and address the concern regarding generalization beyond specific volatility clusters. We will report these results in the rebuttal revision as soon as they are computed.
> - **W2**: We would like to clarify that static features are only utilized at the node level where they provide complementary information, while dynamic features, including price and sentiment, are incorporated through the edge construction process. Therefore, temporal dynamics remain a core component of the model, as the edge formation captures time-dependent relationships and lead–lag interactions across assets. Moreover, the primary focus of the paper is not solely on feature-level temporal signals, but rather on modelling the dynamic topology of the graph, which evolves over time based on the temporal lead–lag structure. This aspect is discussed in Section 3.2 "Dataset collection and graph construction" and in section 4.1 ``Dataset and Metrics''.
> - **W3**: We agree with the reviewer that conducting an in-depth comparison with traditional econometric baselines, such as Granger causality, could enhance the rationale of our work. However, as outlined in Section 3.1 of the article, this comparison poses significant challenges. The reason is that the proposed approach, which models prediction as a link prediction problem in a temporal graph, fundamentally differs from traditional statistical methods. This divergence makes direct comparisons complex and not straightforward. Indeed, adapting econometric baselines, like Granger causality, leads to the development of hybrid approaches that will be inspired by the original methods yet differ from them in key aspects.
>
> Here, instead, we reported the answers to the questions of the reviewer:
> - **Q1**: Please refer to the previously reported response indicated as **W2**.
> - **Q2**: We appreciate the reviewer raising this question, although it's unclear what specific context prompted it. Generally, our results suggest that GraphMixer outperforms other methods, particularly in temporal graphs; the opposite of what is highlighted in the question. Could you provide more details on this point?

---

### Note · Authors · 2026-01-14

I have read and agree with the venue's withdrawal policy on behalf of myself and my co-authors.